# Lipoarabinomannan mediates localized cell wall integrity during division in mycobacteria

Ian L. Sparks[1], Takehiro Kado [1], Malavika Prithviraj[1], Japinder Nijjer[2,3], Jing Yan [2,3] & Yasu S. Morita [1] ✉

The growth and division of mycobacteria, which include clinically relevant pathogens, deviate from that of canonical bacterial models. Despite their Gram-positive ancestry, mycobacteria synthesize and elongate a diderm envelope asymmetrically from the poles, with the old pole elongating more robustly than the new pole. The phosphatidylinositol-anchored lipoglycans lipomannan (LM) and lipoarabinomannan (LAM) are cell envelope components critical for host-pathogen interactions, but their physiological functions in mycobacteria remained elusive. In this work, using biosynthetic mutants of these lipoglycans, we examine their roles in maintaining cell envelope integrity in *Mycobacterium smegmatis* and *Mycobacterium tuberculosis*. We find that mutants defective in producing mature LAM fail to maintain rod cell shape specifically at the new pole and para-septal regions whereas a mutant that produces a larger LAM becomes multi-septated. Therefore, LAM plays critical and distinct roles at subcellular locations associated with division in mycobacteria, including maintenance of local cell wall integrity and septal placement.

Cell envelopes define the shape, growth characteristics, and environmental interactions of bacteria and are broadly classified as Gram-positive or Gram-negative type based on their genetically encoded envelope architecture. *Mycobacterium*, a medically important genus of actinobacteria, produces an envelope divergent from both of these regimes and consequently has drastically different envelope physiology. The mechanisms that govern mycobacterial cell envelope assembly are only beginning to emerge.

Like most other bacteria, mycobacteria have a plasma membrane and a peptidoglycan cell wall. However, mycobacteria also have an arabinogalactan layer covalently attached to the peptidoglycan, which is also bound to long-chain fatty acids called mycolic acids[1-3]. These bound mycolic acids support an outer "myco" membrane composed of free mycolic acids and other lipids. Altogether, this envelope architecture is lipid-dense, and includes multiple hydrophobic and hydrophilic layers, making it difficult for antibiotics to reach their targets. The structural

divergence of mycobacterial envelopes also extends to the mechanisms used by mycobacteria to synthesize these structures during growth and division. Mycobacteria grow asymmetrically from the poles, with the old pole elongating faster than the new pole formed from the most recent cell septation event[4-7]. While we have a basic understanding of the order of events during growth and cell division, as well as genes that are involved in the biosynthesis of each component, many of the structural components of the cell envelope have poorly understood physiological functions. Examples include the phosphatidylinositol mannosides (PIMs) and their derivative lipoglycans lipomannan (LM) and lipoarabinomannan (LAM), which are phosphatidylinositol (PI)-anchored membrane components produced by all mycobacteria[1].

PIM biosynthesis begins on the inner leaflet of the plasma membrane where the mannosyltransferases PimA and PimB' sequentially transfer two mannoses from GDP-mannose to the inositol of PI to produce PIM2 (Fig. 1a)[8-10]. PIM2 is then acylated on one of the mannose

[1]Department of Microbiology, University of Massachusetts, Amherst, MA, USA. [2]Department of Molecular, Cellular and Developmental Biology, Yale University, New Haven, CT, USA. [3]Quantitative Biology Institute, Yale University, New Haven, CT, USA. ✉e-mail: ymorita@umass.edu

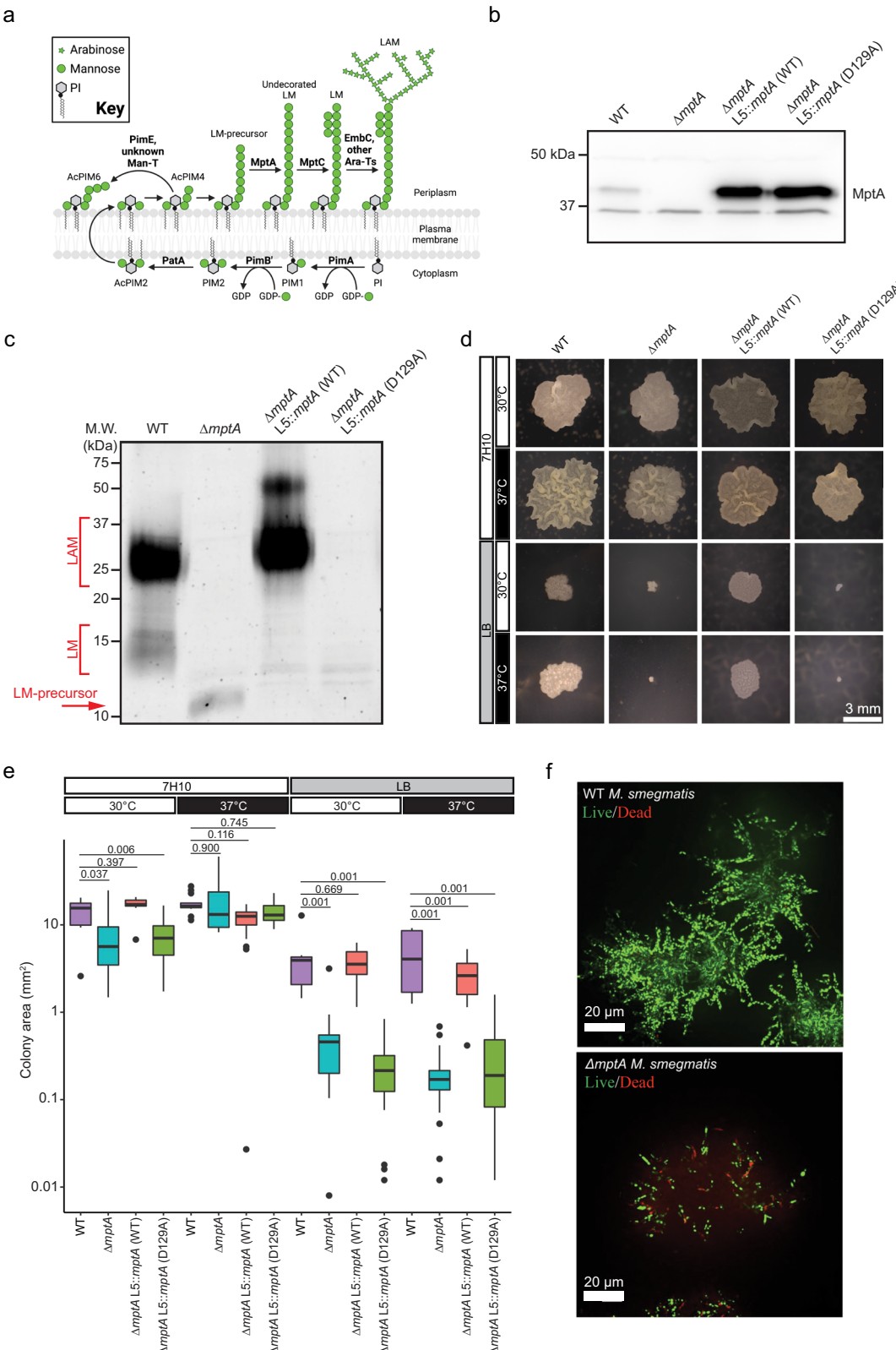

residues, and further mannosylated by unknown transferases to produce a tetra-mannosylated glycolipid (AcPIM4). The downstream fate of AcPIM4 is forked: AcPIM4 may either be mannosylated by the α1-2 mannosyltransferase PimE and another uncharacterized mannosyltransferase to form AcPIM6[11], or AcPIM4 is further mannosylated by α1-6 mannosyltransferases such as MptA to produce LM, a lipoglycan similar to PIMs but with an elongated α1-6 mannose backbone[12,13].

Decorating the long mannan backbone of LM are multiple α1-2-linked mannoses added by the mannosyltransferase MptC[14–16]. Finally, a branched arabinan domain may be added to the α1-6 mannan backbone to produce LAM[1,2]. The biosynthesis of the arabinan domain is carried out by a suite of arabinosyltransferases, including EmbC, as well as some of the Aft transferases, though a detailed understanding of the involvement of each arabinosyltransferase is lacking.

**Fig. 1 | Medium-dependent growth defects of ΔmptA. a** Proposed pathway for PIM, LM, and LAM biosynthesis. Bold text, enzymes. Ara-T, arabinosyltransferase; Man-T, mannosyltransferase. Cartoon made with BioRender. **b** Anti-MptA immunoblot of cell lysates prepared from WT, ΔmptA, ΔmptA L5::mptA (WT), and ΔmptA L5::mptA (D129A) cells. Blot is representative of two independent experiments with similar results. **c** Lipoglycans extracted from WT, ΔmptA, ΔmptA L5::mptA (WT), and ΔmptA L5::mptA (D129A) cells, visualized by glycan staining. Gel is representative of two independent experiments with similar results. **d** Representative example images comparing colony size between WT, ΔmptA, ΔmptA L5::mptA (WT), and ΔmptA L5::mptA (D129A) on Middlebrook 7H10 and LB solid agar media grown at 30 °C or 37 °C. **e** Quantification of colony area for each condition described in Fig. 1d. n = 8, 22, 14, 53 (7H10 30 °C); 15, 15, 21, 9 (7H10 37 °C); 7, 16, 23, 26 (LB 30 °C); 9, 34, 22, 32 (LB 37 °C) colonies for WT, ΔmptA, ΔmptA L5::mptA (WT), and ΔmptA L5::mptA (D129A), respectively. A box spanning the interquartile range (IQR) is drawn from the first quartile to the third quartile with a horizontal line indicating the median. The whiskers extend from the box to the farthest data point within 1.5x IQR from the box. Dots beyond the whiskers indicate potential outliers. Statistical significance was determined by one-way ANOVA and Tukey post-hoc test. **f** Cross-sectional imaging of microcolonies from WT and ΔmptA strains stained with SYTO9 (live, green) and propidium iodide (dead, red) grown as micro-aggregates in LB medium. Images are representative of results obtained from three independent experiments with similar results. Source data for panels **b**, **c**, and **e** are provided in the Source Data file. Colony image data for panels **d** and **e** are available at https://github.com/IanLairdSparks/Sparks_2023. Raw micro-aggregate data images for panel **f** can be accessed via https://doi.org/10.5061/dryad.1vhhmgr1w[72].

PIMs, LM, and LAM specifically bind host receptors during infection to initiate both pro- and anti-inflammatory responses, as well as to prevent phagosome maturation, ultimately promoting intracellular survival (see reviews[17,18]). Notably, a recent study identified LamH, a mannosidase that releases LAM from the cell surface to produce capsular arabinomannan, and is crucial for the survival of *M. tuberculosis* in macrophages[19]. These studies indicate that PIMs, LM, and LAM are important virulence factors during mycobacterial infection. Moreover, these molecules are widely conserved among all mycobacteria, including many non-pathogenic species, suggesting a more fundamental physiological role in the cell envelope[20]. A mannose auxotroph of the nonpathogenic *Mycobacterium smegmatis* becomes shorter and multiseptated when starved for mannose, indicating the physiological importance of mannose-containing molecules such as PIMs, LM, and LAM in cell growth and division[21]. Consistently, deficiencies in LM and LAM biosynthesis have demonstrated loss-of-fitness phenotypes in axenic culture. For example, overexpression of *mptC* in *M. smegmatis* resulted in the truncation of the arabinan and mannan domains of LM and LAM and similar changes in the apparent sizes of LM and LAM are observed in *M. tuberculosis*[16]. In both *M. smegmatis* and *M. tuberculosis*, *mptC* overexpression resulted in increased sensitivity to cell wall-targeting antibiotics[22]. An *M. smegmatis mptA* deletion mutant producing a small LM-precursor but no mature LM or LAM, grew slowly on solid LB agar medium and failed to grow at 42 °C[13]. These observations reveal that LM and LAM may play an important role in mycobacterial growth and cell envelope integrity. Whether the phenotypes described in the literature share an etiological mechanism and what that mechanism might be remains unknown.

In this work, to determine the physiological functions of mycobacterial lipoglycans, we generate various mutants of *M. smegmatis* and *M. tuberculosis* that are deficient in LM and/or LAM biosynthesis and demonstrate that the arabinan domain of LAM defines the subcellularly localized role of LAM in maintaining cell wall integrity during cell division.

## Results

### Defective colony growth of ΔmptA is dependent on culture medium

An *mptA* deletion strain of *M. smegmatis* has a growth defect when grown on solid LB agar at 37 °C[13]. This contrasts with our previous observation that an *mptA* knockdown (KD) strain grows at a rate comparable to wildtype (WT) in Middlebrook 7H9 broth incubated at a lower temperature (30 °C)[22], and suggests conditional importance of LM and LAM for mycobacterial survival and growth. To better understand the role of MptA and its products LM and LAM, we generated ΔmptA and confirmed that MptA was not detectable by immunoblotting in ΔmptA (Fig. 1b). Furthermore, ΔmptA did not produce LM or LAM, but instead accumulated a small LM precursor as reported previously[13,22] (Fig. 1c). We then grew ΔmptA on two different agar plates, LB or Middlebrook 7H10. As previously observed[13], ΔmptA grown on LB agar formed colonies significantly smaller than WT

regardless of the growth temperature (Fig. 1d, e). In contrast, ΔmptA formed colonies that are comparable in size to the WT colonies when grown on Middlebrook 7H10 at either temperature (Fig. 1d, e). These observations suggested that the growth defect of ΔmptA is medium-dependent. We complemented the mutant with an integrative vector carrying a native promoter region fused with either WT *mptA* gene or catalytically inactive D129A *mptA* (see Materials and Methods). The vector was integrated into the mycobacteriophage L5 *attB* site on the chromosome of ΔmptA. Aspartic acid residues in the predicted periplasmic loop between the third and fourth transmembrane domains are suggested to be critical for enzymatic activity[13]. We mutated one of the aspartic acid residues and found that this mutant failed to restore LAM synthesis (Fig. 1c). The lack of phenotypic complementation is not due to the lack of protein production as both WT and D129A mutant MptA were present at comparable levels (Fig. 1b). The levels of MptA production were substantially higher than the endogenous level despite our attempt to use the putative native promoter region to express the gene (Fig. 1b). Interestingly, the WT MptA restored LAM production, but LM failed to accumulate (Fig. 1c), possibly due to the overexpression of *mptA* (see discussion). Nevertheless, the small colony defect of the mutant grown on LB was restored in the complemented strain, ΔmptA L5::mptA (WT), but not with ΔmptA L5::mptA (D129A) (Fig. 1d, e), indicating that the restoration of LAM synthesis is sufficient for the restoration of the colony morphology defect. To examine the growth defect further, we conducted live/dead staining of *M. smegmatis* micro-colonies grown in LB in a glass-bottomed 96-well plate. WT and ΔmptA strains were stained with SYTO9 and propidium iodide such that live cells stained green and dead cells stained red. We found that WT micro-colonies had very few propidium iodide-positive dead cells (Fig. 1f). In contrast, ΔmptA micro-colonies revealed many propidium iodide-stained cells, indicating ΔmptA is more prone to cell death when grown as micro-colonies in LB broth (Fig. 1f). Collectively these results suggest that MptA and its biosynthetic product LAM contribute to cell survival during aggregated growth conditions where LB is used as the medium.

### ΔmptA cannot maintain cell shape and lyses in pellicle growth

We examined pellicle growth as it is an experimentally more tractable and scalable biofilm growth model than colony growth. We compared growth of the WT, ΔmptA, and complemented strains in M63 broth as LB did not support robust pellicle growth of *M. smegmatis*[23]. Because we observed cell death in LB-grown micro-colonies, we examined if ΔmptA lyses under pellicle growth. The pellicle of ΔmptA and its complemented strains showed typical rugose appearance, which was similar to the appearance of the WT pellicle (Supplementary Fig. S1a). However, ΔmptA accumulated significantly higher levels of proteins in their culture medium than the WT (Supplementary Fig. S1b). Mpa is a cytoplasmic protein, which is normally detected at a low level in the spent medium but was detected at a substantially higher level in the spent medium of ΔmptA (Supplementary Fig. S1b). These mutant phenotypes were restored by expressing the WT but not D129A mutant

*mptA* (Supplementary Fig. S1b). Consistent with potential cell lysis, microscopic examination revealed morphological defects of Δ*mptA* (Supplementary Fig. S1c). We quantified morphological defects by measuring the cell width profile across the normalized cell length (Supplementary Fig. S1c) and determining the distribution of maximum cell widths for each strain (Supplementary Fig. S1d). These quantitative analyses revealed that Δ*mptA* shows statistically significant morphological defects, which were restored by WT MptA, but not by D129A mutant MptA (Supplementary Fig. S1c, d). Additionally, we quantified the cell length distributions for WT, Δ*mptA*, and complemented strains and found no significant difference in length between WT and Δ*mptA* cells grown as pellicles in M63 (Supplementary Fig. S1e). Both complemented strains exhibited slight but statistically significant decreases in cell length compared to WT cells (Supplementary Fig. S1e).

## Medium-dependent growth defect of ΔmptA in planktonic culture

The above results suggest that Δ*mptA* and Δ*mptA* L5::*mptA* (D129A) struggle to maintain cell shape and lyse in aggregated growth. While our previous study showed that *mptA* KD cells display no defect when growing planktonically in Middlebrook 7H9[22], we wondered if planktonically grown cells also show similar morphological defects in a medium-dependent manner. When we grew Δ*mptA* in Middlebrook 7H9 broth, they grew similarly to the WT, as previously observed for *mptA* KD cells (Fig. 2a–c). The cell morphology of Δ*mptA* was also comparable to WT, with only a minor increase in cell widths and a similarly minor decrease in cell lengths (Fig. 2d–f). In contrast, when Δ*mptA* was grown in LB broth, there was a growth defect, and its morphology was aberrant, as observed in pellicle growth (Fig. 2g–l). The statistically significant difference in doubling time between WT and Δ*mptA* grown in LB (Fig. 2h) should not be influenced by the morphological defects of Δ*mptA* since doubling time is the rate of growth calculated from the change in optical density of each individual bacterial strain. Nevertheless, we further confirmed that the growth defect in LB measured by optical density at 24 h post inoculation also corresponded to a significant difference in CFU/mL between WT and Δ*mptA* (Fig. 2i), further confirming that planktonic growth of Δ*mptA* is defective in LB. The growth and morphological defects of Δ*mptA* were restored by complementation with WT *mptA* but not with D129A mutant *mptA* (Fig. 2g–k). In LB growth medium, we did not observe any differences in cell length between WT and Δ*mptA* (Fig. 2l). We finally tested osmo-protective LB, in which the medium was supplemented with sorbitol, a non-metabolizable osmolyte. Sorbitol was unable to support the growth of *M. smegmatis* in M9 Minimal Medium as a sole carbon source (Supplementary Fig. S2a). When supplemented in LB at 500 mM, sorbitol suppressed the morphological defects (Supplementary Fig. S2b, c), implying that cell shape deformation is dependent on the osmolarity of the medium. These data together suggest that the growth of Δ*mptA* is conditionally defective in a medium-dependent manner and is not specific to aggregated growth.

## ΔmptA is hypersensitive to beta-lactam antibiotics

The suppression of the morphological defect when Δ*mptA* cells were grown in high-osmolarity growth media (see Supplementary Fig. S2b, c) suggests that the turgor pressure is the force responsible for deforming the Δ*mptA* cell envelope and that the load-bearing function of the cell wall is compromised in the absence of LM and LAM. Previous work showed that *mptC* overexpression (OE), which stunts lipoglycan size, led to increased sensitivity to cell envelope-targeting antibiotics[16,22], further suggesting that diminutive lipoglycans may compromise envelope integrity. Since Δ*mptA* produces even less developed lipoglycans than *mptC* OE, we tested whether this strain is similarly sensitive to cell envelope-targeting antibiotics. We

determined the MIC of seven different drugs for WT and Δ*mptA M. smegmatis* grown in Middlebrook 7H9. We found that Δ*mptA* is over 40 times more sensitive to ampicillin+sulbactam and 8 times more sensitive to meropenem+sulbactam than WT (Supplementary Table S1). Increased sensitivity of Δ*mptA* to the non-beta-lactam drugs was minor in comparison, indicating a specific sensitivity towards drugs targeting peptidoglycan crosslinking in cells deficient in LM and LAM. Sulbactam was added to inhibit the action of beta-lactamase on beta-lactams, but was also added to the non-beta-lactam antibiotic treatments in order to test all drugs under the same condition. Since the morphological defects were detected in LB broth, we suspected that the increased sensitivities of Δ*mptA* to beta-lactams may become even more pronounced in this medium. Indeed, MICs of ampicillin+sulbactam and meropenem+sulbactum for WT became 85 and 128 times that of Δ*mptA*, respectively, demonstrating enhanced beta-lactam sensitivity of Δ*mptA* in LB compared to Middlebrook 7H9 (Supplementary Table S2). Furthermore, supplementing sorbitol reduced the vulnerabilities of Δ*mptA* to these drugs in LB medium (Supplementary Table S2). The increased beta-lactam sensitivities were partially restored in Δ*mptA* complemented with the WT but not with D129A mutant *mptA* (Supplementary Table S3). These observations reiterate the loadbearing defects of Δ*mptA* cell wall that would make the mutant vulnerable to drugs targeting peptidoglycan crosslinking.

## LAM-deficient cells fail to maintain cell shape

Our results suggest that the absence of LM and LAM weakens the structural integrity of the peptidoglycan cell wall, and makes the cells vulnerable to turgor pressure, resulting in characteristic "blebbing" of the cell, observed in other cell wall-defective mutants[24–26]. The complemented strain, Δ*mptA* L5::*mptA* (WT), restored LAM but not LM, implying that LM does not contribute to the morphological defects (see Fig. 1). To provide further insights on the specific structural component of LM/LAM that is essential for maintaining cell shape, we used a collection of other *M. smegmatis* mutants known to produce altered lipoglycans. We first confirmed the expected lipoglycan profiles of each strain under our experimental conditions. Using planktonic LB culture, we harvested log phase cells, extracted their LM and LAM, and visualized LM and LAM by SDS-PAGE (Fig. 3a). Δ*mptC* produces LAM but does not accumulate LM, as previously demonstrated[16]. Having a similar LM/LAM phenotype to Δ*mptA* L5::*mptA* (WT), Δ*mptC* maintained rod shape without any morphological defects (Fig. 3b, c). These results indicate that LM is not critical for maintaining envelope integrity and suggest that LAM is an important lipoglycan for envelope integrity. To further investigate, we analyzed the cell width of an *mptC* OE strain previously shown to produce LM and LAM with both dwarfed mannan and arabinan domains[16]. We reasoned that this strain should display an intermediate morphology defect since it produces a LAM molecule with a small arabinan domain[16]. As expected, *mptC* OE had a significant blebbing defect, but with only 36% of cells in contrast to 67% of Δ*mptA* cells (Fig. 3b, c, compared with Fig. 2j, k). These results suggest that the presence and correct size of the arabinan domain are both required for optimal cell wall integrity. To test this more directly, we initially attempted to delete the gene encoding the key arabinosyltransferase EmbC. Whereas *embC* has been deleted in previous studies[27,28], we were unable to delete the gene in *M. smegmatis* for unknown reasons (see Methods). We therefore knocked down the expression of *embC* by anhydrotetracycline (ATC)-inducible CRISPRi. Being consistent with the phenotype of *embC* deletion mutant reported in previous studies[27,28], *embC* knockdown resulted in a reduced amount of LAM upon ATC addition (Fig. 3a). When this strain was grown planktonically in LB with ATC, 50% of cells displayed a blebbed morphology like that seen for Δ*mptA* and *mptC* OE strains. The morphological defects were observed only when ATC was added. These data demonstrate the importance of the arabinan domain of LAM in maintaining cell shape (Fig. 3b, c). Analysis of cell lengths revealed no

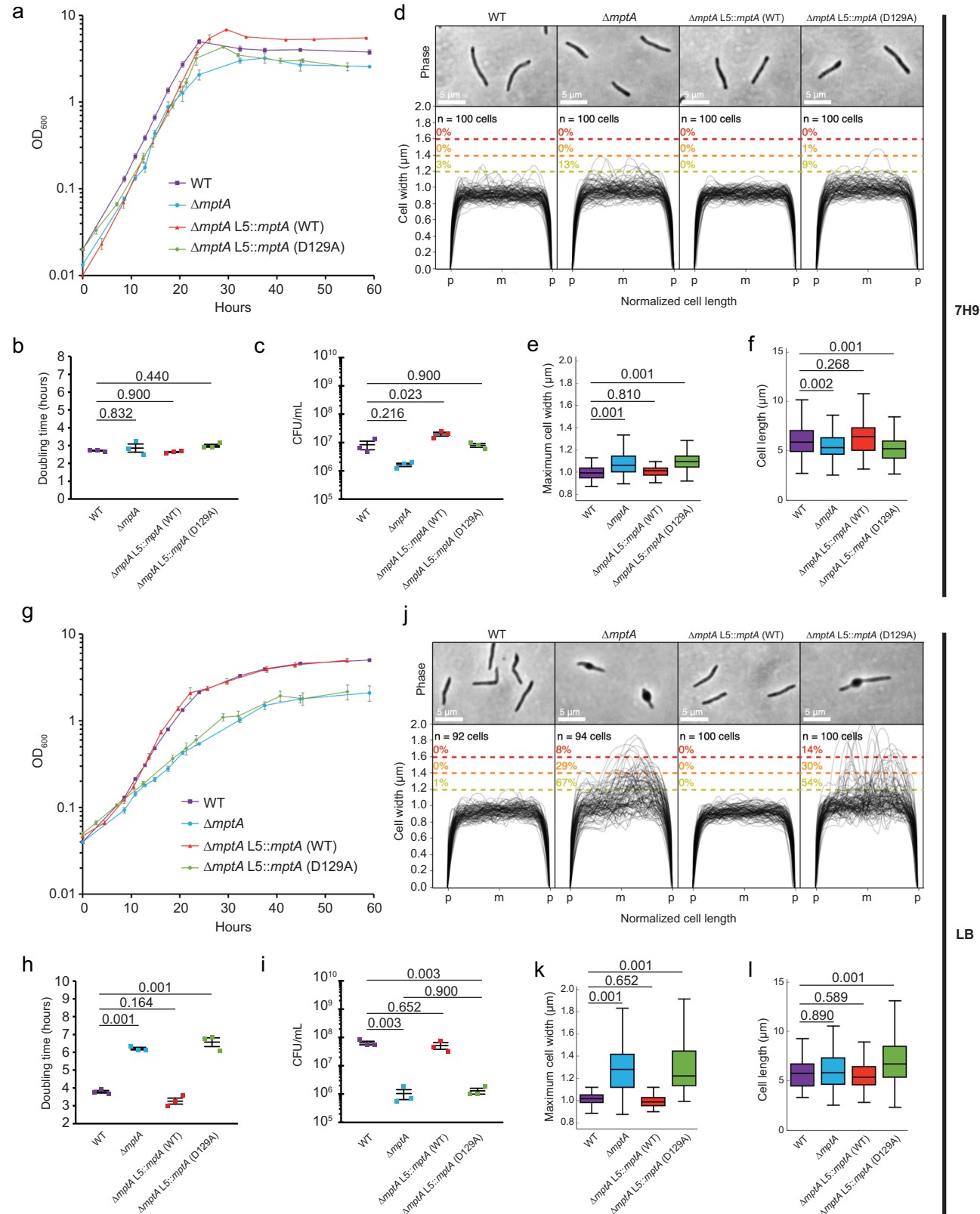

substantial defects, though the mutants were all slightly shorter than WT on average (Fig. 3d).

**Envelope deformations are associated with new poles and septa**

We noticed that many of the cell wall deformations were polar, potentially indicating that the cell wall defect is associated with elongation or division. Due to mycobacteria's asymmetric growth, we can determine whether the blebs are associated with the old pole (cell elongation) or the new pole (septation and division). To determine cell polarity, we incubated cells with a fluorescent D-amino acid (FDAA) analog, called RADA, which is actively incorporated into growing or remodeling cell walls by endogenous cross-linking enzymes[29]. Bright

**Fig. 2 | Growth and cell morphology of planktonically growing cells. a, g** Growth curves of WT, Δ*mptA*, Δ*mptA* L5::*mptA* (WT), and Δ*mptA* L5::*mptA* (D129A) cells grown planktonically in Middlebrook 7H9 (**a**) or LB (**g**) medium. OD$_{600}$, optical density at 600 nm. Error bars represent the standard error of the mean of three biological replicates. **b, h** Doubling times, determined from optical density growth curves, of WT, Δ*mptA*, Δ*mptA* L5::*mptA* (WT), and Δ*mptA* L5::*mptA* (D129A) cells in 7H9 (**b**) or LB (**h**) during log-phase growth. The middle line indicates the mean of three biological replicates (indicated with colored squares). The bottom and top lines represent the standard error of the mean. **c, i** CFU/mL of WT, Δ*mptA*, Δ*mptA* L5::*mptA* (WT), and Δ*mptA* L5::*mptA* (D129A) cells in 7H9 (**c**) or LB (**i**) at the 24-h timepoint. The middle line indicates the mean of 3 biological replicates (indicated with colored squares). The bottom and top lines represent the standard error of the mean. **d, j** Phase contrast micrographs and cell width profiles of planktonic WT, Δ*mptA*, Δ*mptA* L5::*mptA* (WT), and Δ*mptA* L5::*mptA* (D129A) cells grown in 7H9 (**d**) and LB (**j**). Each cell's length was normalized to the same length with "p" and "m" indicating cell poles and mid-cell, respectively. The percentage values above the

dotted colored lines indicate the portion of cells exhibiting maximum cell widths greater than or equal to the corresponding cell width threshold. **e, k** Boxplots comparing the distribution of maximum cell widths between WT, Δ*mptA*, Δ*mptA* L5::*mptA* (WT), and Δ*mptA* L5::*mptA* (D129A) strains grown planktonically in 7H9 (**e**) or LB (**k**). *n* = 100 cells for each strain in panel (**e**) and *n* = 92, 94, 100, 100 cells for WT, Δ*mptA*, Δ*mptA* L5::*mptA* (WT), and Δ*mptA* L5::*mptA* (D129A), respectively in panel (**k**). **f, l** Boxplots comparing the distribution of cell lengths between WT, Δ*mptA*, Δ*mptA* L5::*mptA* (WT), and Δ*mptA* L5::*mptA* (D129A) strains grown plank-tonically in 7H9 (**f**) or LB (**l**). The boxplots are drawn as described in Fig. 1. Statistical significance for each experiment was determined by one-way ANOVA and Tukey post-hoc test. *n* = 174, 162, 111, 136 cells for WT, Δ*mptA*, Δ*mptA* L5::*mptA* (WT), and Δ*mptA* L5::*mptA* (D129A) respectively in panel (**f**). *n* = 92, 94, 121, 158 cells for WT, Δ*mptA*, Δ*mptA* L5::*mptA* (WT), and Δ*mptA* L5::*mptA* (D129A), respectively in panel (**l**). Source data for each sub-panel are provided in the Source Data file. Microscopy image data for panels **d–f**, **j–l** are available at https://github.com/IanLairdSparks/Sparks_2023.

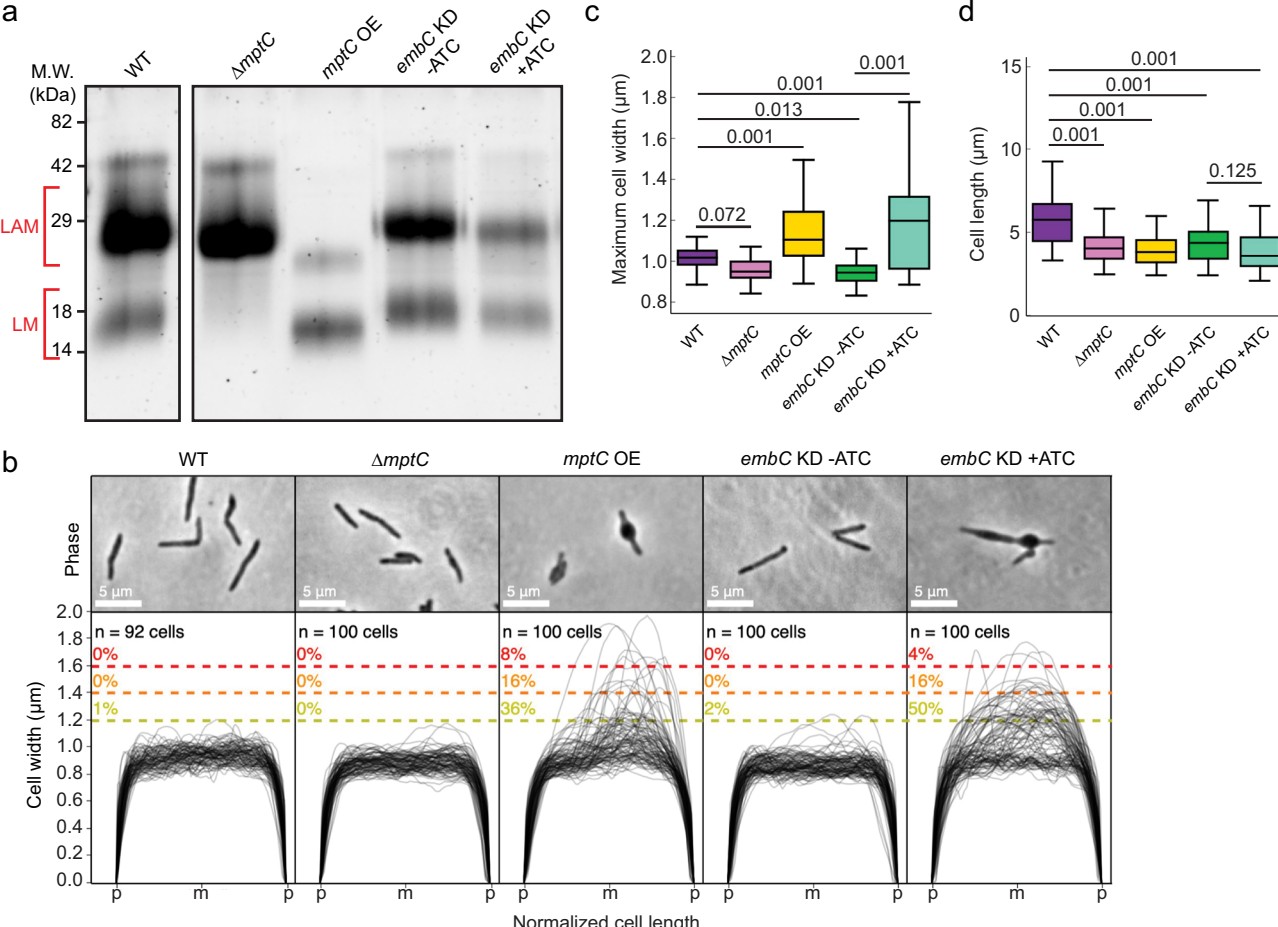

**Fig. 3 | Cells deficient in LAM fail to maintain cell shape. a** SDS-PAGE of lipo-glycans extracted from WT and various lipoglycan mutants, visualized by glycan staining (WT and mutant lanes are from the same PAGE gel). *mptC* OE, *mptC* overexpression strain; *embC* KD, *embC* CRISPRi knockdown strain. ATC, anhy-drotetracycline. Gel is representative of two independent experiments with similar results. **b** Phase contrast micrographs and cell width profiles for each strain when grown planktonically in LB medium. Cell width profiles were graphed as described in Fig. 2. "p" and "m" indicate cell poles and mid-cell, respectively. **c, d** Boxplots

comparing the distribution of maximum cell widths (**c**) and cell lengths (**d**) among mutant strains. *n* = 92, 100, 100, 100 (panel **c**); 92, 201, 119, 102, 187 (panel **d**) cells for WT, Δ*mptC*, *mptC* OE, *embC* KD -ATC, and *embC* KD +ATC, respectively. The boxplots are drawn as described in Fig. 1. Statistical significance was determined by one-way ANOVA and Tukey post-hoc test. Source data for each sub-panel are provided in the Source Data file. Microscopy image data for panels **b–d** are avail-able at https://github.com/IanLairdSparks/Sparks_2023.

and dim RADA labeling of cell poles correlates with old and new poles, respectively[24]. As expected, WT and Δ*mptA* cells grown in LB plank-tonic culture were labeled asymmetrically at the cell poles, with one pole labeling brighter and extending farther into the sidewall,

representing the old pole, and one pole labeling dimmer, representing the new pole. We then aligned each non-septated blebbed Δ*mptA* cell according to old/new pole determined by its RADA labeling and plot-ted the cell width profiles. The majority of maximum cell widths for

non-septated blebbed cells associated with new poles, though some blebs were associated with the midcell (Supplementary Fig. S3a). Intriguingly, ΔmptA cells also displayed higher RADA labeling at the sidewall, particularly at deformed regions of the cell envelope on the new pole half of the cell, indicating increased peptidoglycan remodeling along the sections of the cell envelope that are not actively growing (Supplementary Fig. S3b, c). The sidewall RADA labeling was not observed once ΔmptA was complemented with the expression of WT mptA but was observed when complemented with D129A mutant mptA (Supplementary Fig. S3b, c). The increased FDAA labeling at the regions of envelope deformations suggests that intense remodeling was induced to repair damaged or weak peptidoglycan cell wall. Since RADA labels septa, we also noticed that cell envelope deformations in septated ΔmptA cells were often associated with septa, further suggesting a link between division and LAM function (Supplementary Fig. S3d). These data suggest that LAM is specifically important for peptidoglycan integrity during or immediately after septation and daughter cell separation and not the active elongation of old poles.

### PonA1 restores envelope integrity in LM/LAM-deficient cells

The new pole/septal location of cell envelope deformation suggests LAM may be necessary for controlling septal peptidoglycan hydrolase activity. Septal hydrolases must be able to cut the peptidoglycan connecting two daughter cells to achieve cell separation without destroying the peptidoglycan associated directly with each daughter cell's inner membrane. This implies that some periplasmic factor spatially restricts the activity of septal hydrolases to avoid indiscriminate cell wall damage around the septum/new pole, and LAM may be involved in such spatial regulations of septal hydrolases. The predominant septal hydrolase in *M. smegmatis* is RipA, which forms a complex with another hydrolase RpfB. The synergistic activity of this septal hydrolase complex is directly inhibited by the penicillin-binding protein PonA1 through a RipA-binding motif on PonA1's C-terminus (Fig. 4a)[25,30–32]. To test whether RipA/RpfB complex inhibition by PonA1 could prevent envelope blebbing, we constitutively overexpressed *ponA1* in ΔmptA, and confirmed the overexpression by Bocillin FL Penicillin labeling (Fig. 4b). Since mptA is deleted, we did not expect that LM/LAM biosynthesis would be restored in ΔmptA L5::ponA1. We confirmed that only a small LM intermediate was detectable in ΔmptA L5::ponA1 (Fig. 4c). Strikingly, ΔmptA L5::ponA1 maintained rod shape cell morphology (Fig. 4d–f) and restored normal RADA labeling (Fig. 4g), indicating that the expression of PonA1 successfully rescued the peptidoglycan defect in the LM/LAM deficient ΔmptA mutant. To directly determine whether the cell wall defect of ΔmptA is dependent on RipA and its downstream operon partner RipB, an ATC-inducible *ripAB* CRISPRi KD construct was integrated into the L5 site of the ΔmptA strain (ΔmptA L5::ripAB KD) and the WT strain (WT L5::ripAB KD). When ATC was not added to the culture medium, each uninduced strain mimicked the cell morphology phenotype of their parental strains: ΔmptA L5::ripAB KD blebbed and WT L5::ripAB KD formed healthy rod-shaped cells (Fig. 4h). Consistent with published results[31,32], induction of *ripAB* knockdown via addition of ATC resulted in cell elongation and ectopic pole formation/branching in both genetic backgrounds, indicative of failed division in the absence of the key septal hydrolase, while the cell width characteristics of each strain remained similar to uninduced conditions (Fig. 4h). Notably, healthy cell width characteristics and peptidoglycan remodeling were not restored in ΔmptA L5::ripAB KD upon knockdown of the key septal hydrolase RipA, indicating that the cell wall defect of ΔmptA is not due to the dysregulation of RipA.

### Cells producing a large LAM are defective in cell separation

Since dwarfed LAM weakens cell wall near the septum, a larger LAM may have the opposite effect of preventing the localized weakening of cell wall, like the *ripA*-deficient mutant, which cannot separate the two

daughter cells and becomes multi-septated. We previously demonstrated that overexpression of *mptA* results in larger LAM molecules[16]. To determine any potential impacts producing a larger LAM molecule has on the cell wall or cell division, we transformed the WT strain with the acetamide-inducible *mptA* expression vector as before[16]. Acetamide-inducible promoter was leaky and MptA was produced in the transformant without acetamide addition at a level higher than the endogenous WT level (Fig. 5a). Upon incubation with acetamide for 8 h, there was a robust overproduction of MptA (Fig. 5a), and the apparent molecular weight of LAM became larger (Fig. 5b). We have previously shown that both mannan and arabinan part of LAM becomes larger by the acetamide-induced MptA overproduction[16]. While larger LAM did not affect the cell width (Fig. 5c), there were some cells that were quite long, although this slight increase in cell length at the population level was not significant compared to the vector control (Fig. 5d). After incubation with acetamide, the cells were labeled with the FDAA peptidoglycan probe HADA, which indicated a hyperseptation phenotype in a subset of cells (Fig. 5e). We quantified the percentage of septated and multiseptated cells in the *mptA* overexpression strains (with or without acetamide addition) and compared with empty vector control, WT, ΔmptA, and ΔmptC to determine the prevalence of this cell division defect at the population level (Fig. 5f). While only 20% of WT and ΔmptA cells were septated of which 2% were multiseptated, 46% of *mptA* OE cells were septated of which 29% were multiseptated upon induction of gene expression by acetamide. Without acetamide-induced *mptA* overexpression, 0% of cells were multi-septated, demonstrating that strong overexpression of *mptA* inhibits the successful resolution of septation events. Since LM does not accumulate in the *mptA* OE strain (Fig. 5b), the division defect could either result from the lack of LM or the production of larger LAM. To test this, we observed the level of septation in the ΔmptC strain, which also does not accumulate LM but does not produce a large LAM molecule (see Fig. 3a). 24% of ΔmptC cells had a single septum and none were multiseptated, indicating that the lack of LM in this strain did not lead to a drastic increase in the number of septa over WT. Together, these results indicate that biosynthesis of an abnormally large LAM, not the lack of LM accumulation, correlates with the division defect.

A previous study found that in *M. tuberculosis*, the protease MarP proteolytically activates the septal peptidoglycan hydrolase RipA in response to low pH[33]. MarP is conserved in *M. smegmatis*. We therefore reasoned that acidic growth conditions, which increase the activity of RipA, may increase septal hydrolysis enough to resolve the multi-septation effect associated with oversized LAM in *mptA* OE. To test this, we grew WT and *mptA* OE in LB adjusted to pH 5.5 and compared the level of septation with that observed when cells were grown in standard LB (pH = 7) (Fig. 5g). The acidic growth condition nullified the hyperseptation phenotype, suggesting that increased RipA activity may rescue this division defect. While increased RipA activity may help cells with large LAM to divide, LAM's function does not appear to be directly involved in modulating RipA activity per se as we observed that knocking down this hydrolase did not rescue cells deficient in LAM (see Fig. 4h). If this is indeed the case, then the division defect observed in *mptA* OE may be phenotypically distinct from that of cells deficient in RipA. In support of this, we found that while both *mptA* OE and induced WT L5::ripAB KD have a multiseptation defect, the spacing between septa is markedly different between the two strains (Fig. 5h). Knocking down *ripAB* results in consistent interseptal distances, indicative of regular cell wall elongation followed by failed cell separation, while *mptA* OE displays much less consistent and shorter interseptal distances. Shorter interseptal distances in *mptA* OE correlated with shorter cell length (Fig. 5i). Multiseptated *ripAB* KD cells are at least ~9 μm long while *mptA* OE cells that are as short as ~5 μm can become multiseptated. These data are consistent with the possibility that *mptA* OE is defective in post-septation cell elongation or prone to uncontrolled hyperseptation. It is worth noting that some septa were so close

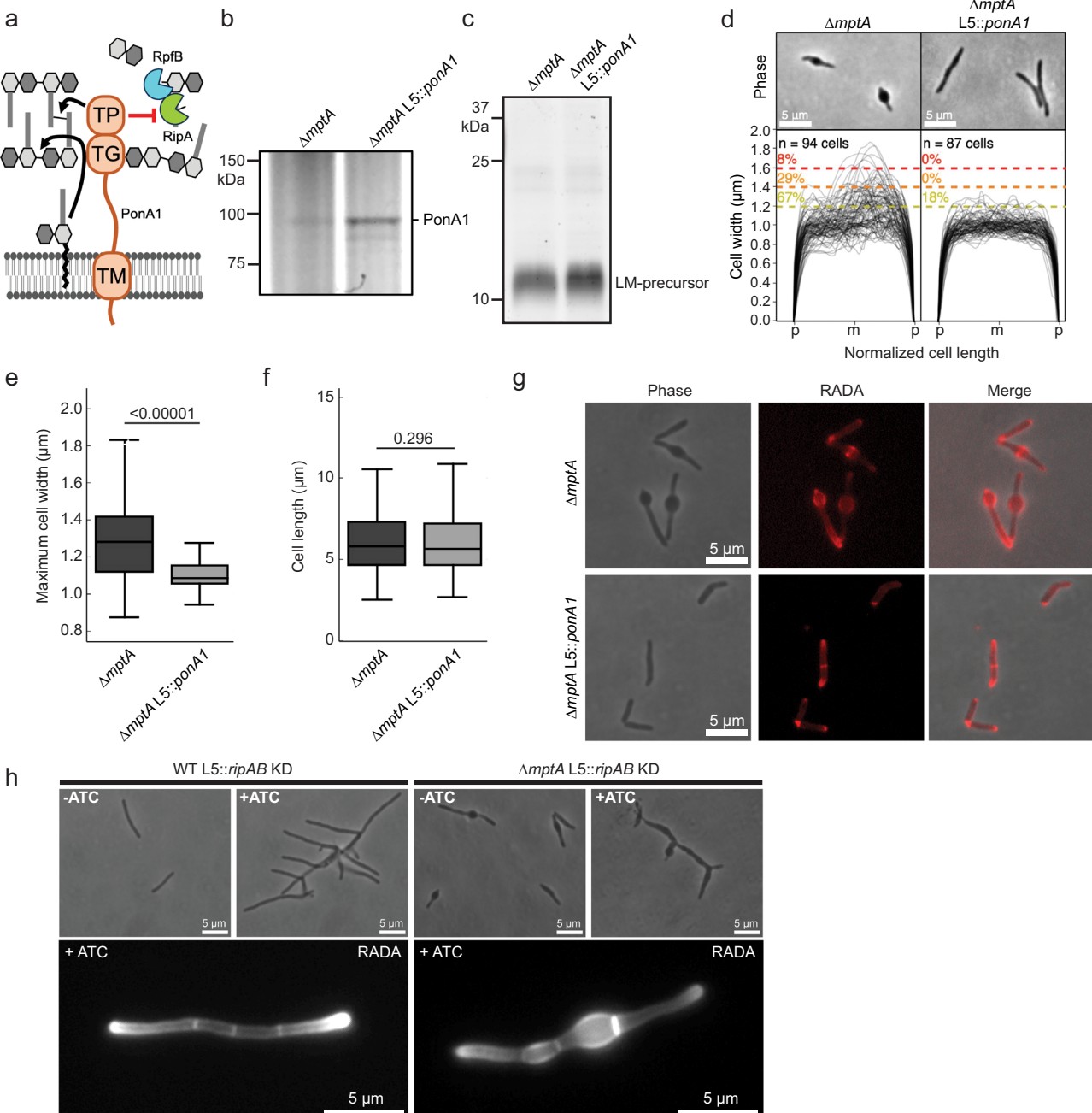

**Fig. 4 | PonA1 rescues the morphological defects of Δ*mptA*. a** The domain structure of PonA1 and the corresponding cell wall substrates and interacting partners. The C-terminal domain of PonA1 interacts with RipA to suppress the activity of the peptidoglycan hydrolase complex formed by RipA and RpfB. TG, transglycosylase domain; TP, transpeptidase domain; TM, transmembrane domain. Cartoon made with Microsoft PowerPoint. **b** SDS-PAGE of Δ*mptA* and Δ*mptA L5::ponA1* cell lysates in which PonA1 is stained with the fluorescent penicillin analog bocillin. Gel is representative of two independent experiments with similar results. **c** SDS-PAGE of lipoglycans extracted from Δ*mptA* and Δ*mptA L5::ponA1*, visualized by glycan staining, showing that *ponA1* overexpression does not restore LM/LAM biosynthesis. Gel is representative of two independent experiments with similar results. **d** Phase contrast micrographs and cell width profiles of Δ*mptA* cells and Δ*mptA* cells overproducing PonA1 (Δ*mptA L5::ponA1*). Cell width profiles were graphed as described in Fig. 2. "p" and "m" indicate cell poles and mid-cell,

respectively. **e, f** Boxplots comparing the distribution of maximum cell widths (**e**) and cell lengths (**f**) between Δ*mptA* and Δ*mptA L5::ponA1*. $n = 94, 87$ cells for Δ*mptA* and Δ*mptA L5::ponA1*, respectively. The boxplots are drawn as described in Fig. 1. Statistical significance was determined using the one-tailed student's T-test. In panel (**e**), $P = 3.58 \times 10^{-13}$. **g** Phase contrast and fluorescence micrographs of RADA-labeled Δ*mptA* and Δ*mptA L5::ponA1* cells. Microscope images are representative of two independent experiments with similar results. **h** Top: phase contrast micrographs of ATC-inducible *ripAB* CRISPRi knockdown strains constructed in the WT and Δ*mptA* genetic backgrounds. *ripAB* KD was induced for 20 h. Bottom: fluorescent micrographs of ATC-inducible *ripAB* KD strains stained with RADA. *ripAB* KD was induced for 8 h. Microscope images are representative of two independent experiments with similar results. Source data for panels **b**–**f** are provided in the Source Data file. Microscopy image data for panels **d**–**h** are available at https://github.com/IanLairdSparks/Sparks_2023.

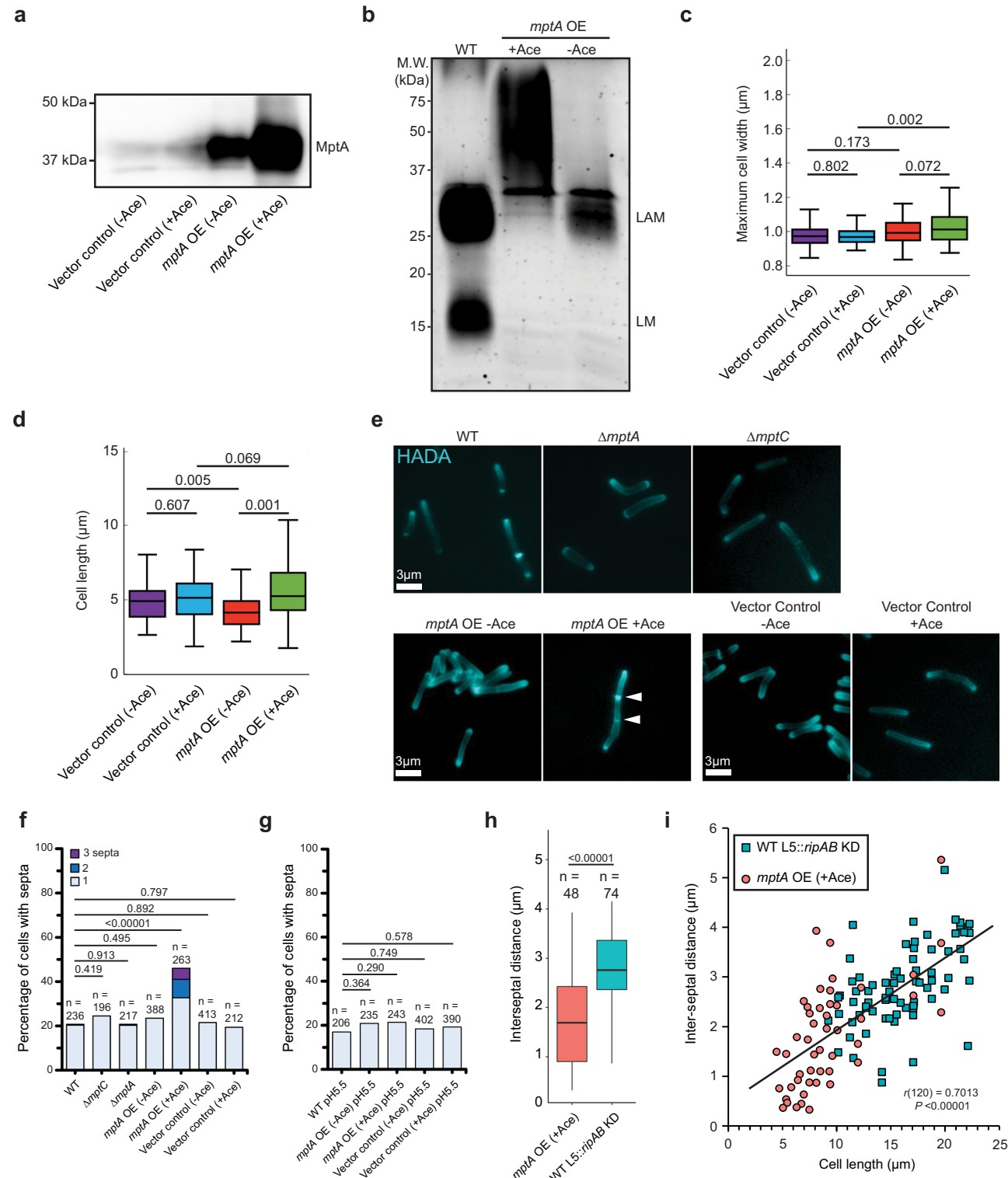

together in the *mptA* OE strain that they may be one "spiral" shaped septum resulting from an improperly condensed Z-ring. Overall, these data support a model in which the size of LAM impacts cell envelope structure at sites of division, leading to the success or failure of cell division and/or subsequent elongation.

## mptA knockdown recapitulates cell shape defects in M. tuberculosis

All genome-wide mutant screening studies predict that *mptA* is essential in *M. tuberculosis*[34–36]. A recent genome-wide CRISPRi KD study also supported the same conclusion, and further indicated that *mptA* KD results in significantly higher sensitivity to vancomycin, a drug targeting cell wall biosynthesis[37]. We created an *mptA* CRISPRi knockdown construct, and the strain transformed with the construct was tested for the predicted growth defect. As shown in Fig. 6a, the *mptA* KD strain showed a growth delay after the second subculturing with ATC. We determined the reduced levels of MptA protein at the end of the second subculture by immunoblotting using anti-MptA antibody (Fig. 6b). Furthermore, the levels of LM and LAM were reduced (Fig. 6c), and the cell morphology became aberrant (Fig. 6d).

**Fig. 5 | Large LAM biosynthesis results in multiseptated cells. a** Anti-MptA immunoblot of cell lysates prepared from *mptA* OE (-/+ acetamide), and vector control (-/+ acetamide) cells. Equal protein amount was loaded per lane. Ace, acetamide. Blot is representative of two independent experiments with similar results. **b** SDS-PAGE of lipoglycans extracted from WT, and the *mptA* OE strain with and without acetamide, visualized by glycan staining. Gel is representative of two independent experiments with similar results. **c, d** Boxplots comparing the distribution of maximum cell widths (**c**) and cell lengths (**d**) of *mptA* OE (-/+ acetamide), and vector control (-/+ acetamide) strains. *n* = 100, 100, 100, 100 (panel **c**) and 103, 105, 161, 118 cells (panel **d**) for vector control - acetamide, vector control + acetamide, *mptA* OE - acetamide, and *mptA* OE + acetamide, respectively. The boxplots are drawn as described in Fig. 1. Statistical significance was determined by one-way ANOVA and Tukey post-hoc test. **e** Fluorescence micrographs of HADA-labeled cells. **f** Septa enumeration for WT and lipoglycan mutant cells grown planktonically in LB. The combined bar height indicates the total percentage of cells with one or more septa. Statistical significance was determined by one-sided chi-square tests, with total non-septated and total septated cell counts as the two categorical variables compared between strains. **g** Septa enumeration for WT, *mptA*

OE (-/+ acetamide), and vector control (-/+ acetamide) strains grown planktonically in acidic (pH 5.5) LB medium. Statistical significance was determined as described in panel (**f**). **h** Comparison between the distribution of septum-to-septum distances observed in multiseptated acetamide-induced *mptA* OE cells and multiseptated WT L5::*ripAB* KD cells. n values represent the total number of septum-to-septum distances measured in multiseptated cells of each strain. *mptA* overexpression and *ripAB* knockdown were both induced for 8 h. The boxplots are drawn as described in Fig. 1. Statistical significance was determined using the one-tailed student's T-test. $P = 1.41 \times 10^{-8}$. **i** Correlation of cell lengths to inter-septal distances, comparing multiseptated acetamide-induced *mptA* OE cells (red circle) and multiseptated WT L5::*ripAB* KD cells (blue square). Pearson correlation analysis was conducted to determine the strength of correlation (agnostic to genetic background); septum-to-septum distance and cell length were found to be moderately positively correlated, $r(120) = 0.70$, $P = 2.36 \times 10^{-19}$. Statistical significance was determined using the two-tailed student's T-test. Source data for panels **a**–**d** and **f**–**i** are provided in the Source Data file. Microscopy image data for panels **c**–**i** are available at https://github.com/IanLairdSparks/Sparks_2023.

Taken together, these data recapitulate the phenotype of *M. smegmatis* Δ*mptA* and support that *mptA* is an essential enzyme in *M. tuberculosis*.

## Overexpression of mptC induces cell shape defects in M. tuberculosis

Since overexpression of *mptC* led to morphological defects in *M. smegmatis* (see Fig. 3), we next overexpressed *mptC* to further examine the role of lipoglycans in *M. tuberculosis* cell shape maintenance. We have previously overexpressed *mptC* in *M. tuberculosis* and showed that the mutant produces a reduced amount of smaller LAM although the apparent size changes were not as drastic as what were observed in *M. smegmatis*[16] (see Supplementary Fig. S4e). *mptC* OE also makes the *M. tuberculosis* mutant more sensitive to beta-lactams[22], suggesting the potential impact of LM/LAM on peptidoglycan integrity. Since we previously showed that a D45A mutation makes *M. smegmatis* MptC catalytically inactive[16], we introduced an equivalent D46A mutation in *M. tuberculosis* MptC and overexpressed *mptC* encoding either the WT or D46A mutant versions in *M. tuberculosis*. The overexpression of WT *mptC*, but not D46A *mptC*, induced morphological defects in *M. tuberculosis* (Supplementary Fig. S4a–d). Similar to *mptC* OE in *M. smegmatis*, maximum cell width became substantially larger when WT *mptC* is overexpressed in *M. tuberculosis* (Supplementary Fig. S4b). Cell length was not affected (Supplementary Fig. S4c). D46A MptC did not induce any morphological defects. Immunoblotting confirmed that both WT MptC and D46A mutant MptC were overexpressed at comparable levels relative to the endogenous level in WT *M. tuberculosis* (Supplementary Fig. S4d). Despite the overexpression, D46A MptC had no effect on LM/LAM profile, in contrast to WT MptC over-expression (Supplementary Fig. S4e), being consistent with the mutant being catalytically inactive. Taken together, these results further support an evolutionarily conserved role of LAM in mycobacterial cell division.

## Discussion

In 1975, Norman Shaw proposed to use the term lipoglycan for Gram-positive macroamphiphiles that are structurally and functionally distinct from Gram-negative lipopolysaccharides[38]. Lipoglycans are widely found in Gram-positive bacteria, including Actinobacteria, that do not produce lipoteichoic acids (LTAs). This intriguing observation led to a speculation that LTAs and lipoglycans fulfill functionally equivalent roles[39–41], but the analogy has been purely speculative. In the current study, we showed that the mycobacterial lipoglycan, LAM, regulates septation and division through modulating cell wall integrity. A similar role in septation and cell division has been ascribed to LTAs in several Gram-positive bacteria[42–45]. In particular, *S. aureus* LTA

biosynthesis proteins localize to division sites and interact directly with multiple divisome and cell wall synthesis proteins[46], and *S. aureus* mutants lacking LTA display aberrant septation and division as well as shape defects associated with cell lysis[44,47]. Furthermore, excessively long LTAs result in defective cell division, leading to cell "chaining" phenotypes[48]. These phenotypes are strikingly similar to what we observed in *M. smegmatis* LAM mutants, suggesting important roles these lipopolymers may play during cell division. Since LAM is not present in many other Gram-positive bacteria that do not produce LTAs, it remains to be tested if other types of lipoglycans play similar roles in other bacteria.

Our results corroborate and unify previous findings suggesting the medium-dependent importance of LAM in mycobacterial growth and envelope integrity. The early report of Δ*mptA*'s small colony phenotype on LB was the first indication that LM and/or LAM were important for mycobacterial viability[13]. In contrast, a later report showed that an *mptA* conditional knockdown strain grew normally in Middlebrook 7H9 growth medium[22]. Our current study verified the cause of the contrasting observations in the previous studies and demonstrated the importance of culture medium in determining the fitness of Δ*mptA*. The role LAM plays in growth is further supported by the observation of a similar small colony phenotype on complex solid medium for an *M. smegmatis lpqW* deletion mutant. LpqW acts at the branch point in PIM/LM/LAM biosynthetic pathway to shunt AcPIM4 more towards LM/LAM biosynthesis than AcPIM6 biosynthesis[49]. Presumably due to the redirection of AcPIM4 to AcPIM6 synthesis, Δ*lpqW* failed to accumulate LM and LAM[49]. Δ*lpqW* accumulates mutations in *pimE*, preventing the conversion of AcPIM4 to AcPIM6, when grown on LB but not on Middlebrook 7H10, resulting in restored colony size and biosynthesis of LM and LAM[50]. Additional studies are needed to identify what makes Δ*mptA* and Δ*lpqW* struggle to grow on LB. It is also notable that WT MIC values for cell wall targeting drugs vary considerably between LB and Middlebrook media, which may reflect significant medium-dependent differences in cell envelope architecture and dynamics that may dictate the importance of mycobacterial lipoglycans.

One clue as to how LM and LAM may be important for growth came from the observation that *mptC* OE truncated LM and LAM and concomitantly increased sensitivity to cell wall-targeting antibiotics[22]. This observation suggested that lipoglycan deficiency may compromise the mycobacterial cell wall in some way. In the current study, we demonstrated that peptidoglycan remodeling, and its load-bearing and shape maintenance functions are compromised in the absence of LM and LAM. Furthermore, we demonstrated that LM/LAM-deficient cells are specifically sensitive to inhibition of cell wall crosslinking, and that cell shape defects and de-localized peptidoglycan remodeling are

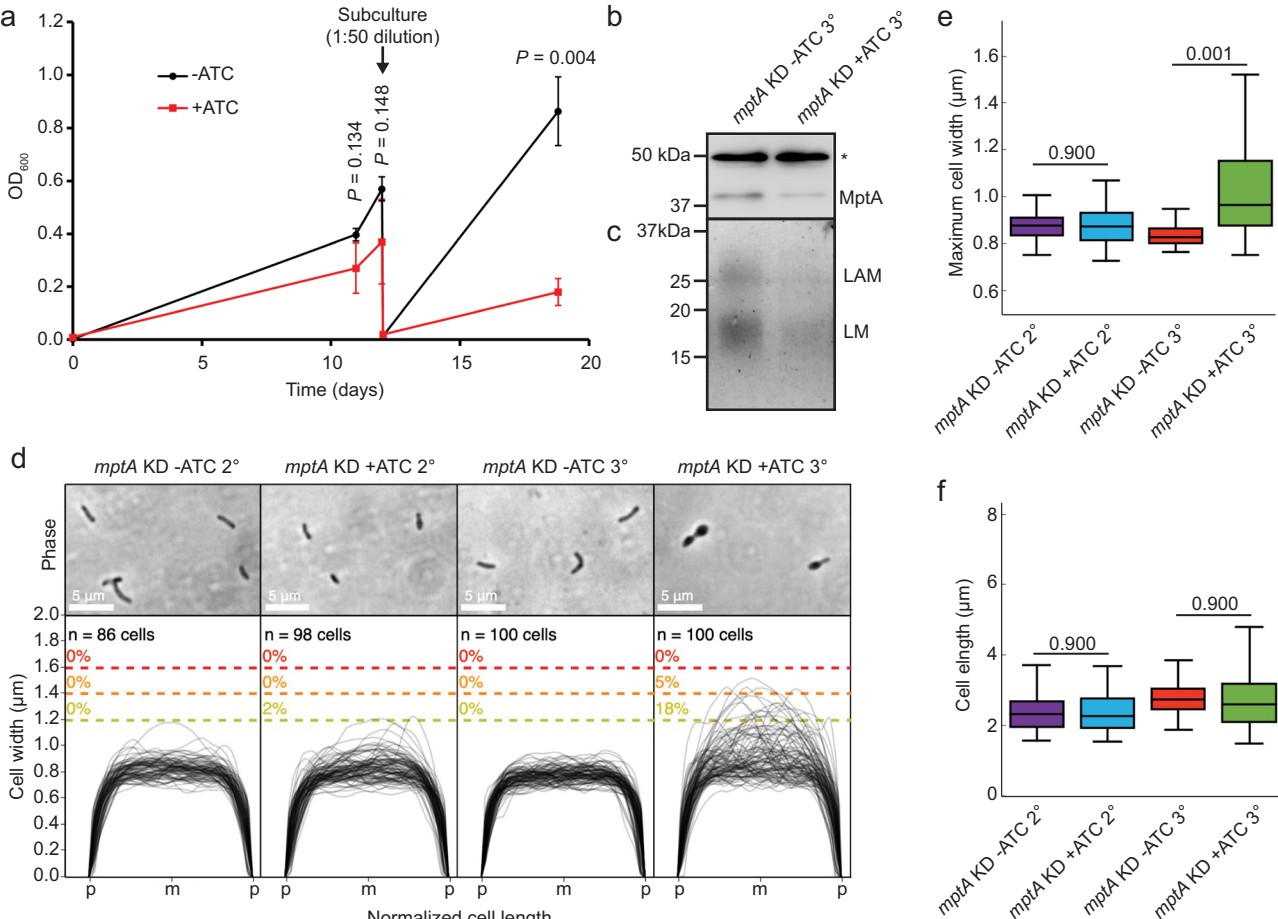

**Fig. 6 | CRISPRi knockdown of *mptA* induces morphological and growth defects in *M. tuberculosis*. a** Growth delay of *M. tuberculosis mptA* KD induced with ATC (red square) compared to the uninduced condition (black circle), measured using optical density at 600 nm. The cells were sub-cultured twice with continuous exposure to 100 ng/mL ATC. The first sub-culture (2°) was sub-cultured into fresh medium (1:50 dilution) on day 12 (arrow) and incubated for an additional seven days (3° culture). Averages of biological triplicates with error bars representing standard errors of the mean are shown. Statistical significance of growth differences between -ATC and +ATC conditions were determined using the one-tailed student's T-test. **b** Anti-MptA immunoblot of *M. tuberculosis* cell lysates prepared from *mptA* KD (-/+ ATC) at the end of the second subcultures (3°). *, a non-specific protein, showing equal loading of two samples. Blot is representative of two independent experiments with similar results. **c** SDS-PAGE of lipoglycans extracted from *M. tuberculosis mptA* KD second subcultures (3°) with and without ATC, visualized by glycan staining. Gel is representative of two independent experiments with similar results. **d** Phase contrast micrographs and cell width profiles for *M. tuberculosis mptA* KD at the end of the first (2°) and second (3°) sub-cultures with and without ATC. Cell width profiles were graphed as described in Fig. 2. "p" and "m" indicate cell poles and mid-cell, respectively. **e, f** Boxplots comparing the distribution of maximum cell widths (**e**) and cell lengths (**f**) between *M. tuberculosis mptA* KD at the end of the first (2°) and second (3°) sub-cultures with and without ATC. The boxplots are drawn as described in Fig. 1. Statistical significance was determined by one-way ANOVA and Tukey post-hoc test. *n* = 86, 98, 100, 100 cells for *mptA* KD - ATC 2°, *mptA* KD + ATC 2°, *mptA* KD - ATC 3°, *mptA* KD + ATC 3°, respectively, in panel **e**. *n* = 86, 98, 102, 165 cells for *mptA* KD - ATC 2°, *mptA* KD + ATC 2°, *mptA* KD - ATC 3°, *mptA* KD + ATC 3°, respectively, in panel **f**. Source data for each sub-panel are provided in the Source Data file. Microscopy image data for panels **d**–**f** are available at https://github.com/IanLairdSparks/Sparks_2023.

subcellularly localized to regions associated with recent division and not general elongation in LM/LAM-deficient cells. By testing the cell morphology characteristics of five distinct LM/LAM biosynthetic mutants, we determined that full-sized LAM, and not LM, is important for maintaining envelope integrity. A recent study indicated that the α1-6 mannan backbone of LM/LAM carries 13-18 mannose residues[51]. Using CHARMM-GUI Glycan Modeler, the chain of 13 α1-6-linked mannoses is predicted to form a 4.2 nm helix[52,53]. It should be noted that intermolecular interactions with other periplasmic structures, such as proteins, could affect the coiling of the α1-6-linked helical mannan of LM and LAM and thus change its length to some degree. Nevertheless, given the predicted height of the periplasmic space being 14-17 nm[54], the mannan backbone part of LM/LAM is probably not sufficient to span the periplasmic space and reach the peptidoglycan layer. However, ~23 residues of α1-5-linked arabinose, found as a branch in the arabinan part of LAM[51], can extend LAM by 9.8 nm,

which makes it long enough to reach the peptidoglycan layer. Multiple branches of these arabinan chains increase the surface area/coverage of the lipoglycan structure precisely where the molecule may interact with the cell wall or cell wall-acting proteins. Our data are consistent with defects in peptidoglycan integrity, but the defects may be present in other parts of the cell envelope. Although it is beyond the scope of the current study, detailed structural analyses are needed to determine the precise structural defects in the cell envelope of Δ*mptA*.

Cells lacking sufficient cell wall hydrolase activity fail to fully divide, resulting in a multi-septation phenotype[32]. We initially hypothesized that LAM could be modulating septal hydrolase activity such that in the absence of LAM, hydrolase activity becomes too active and when LAM is too big, hydrolase activity is impaired. However, we determined that the key septal hydrolase, RipA, was not responsible for the cell shape defect observed in LAM-deficient cells, as knocking down *ripAB* expression in the Δ*mptA* background did not relieve cell

shape defects. The compounded phenotypes of *mptA* deletion and *ripAB* knockdown demonstrate that the cell wall defects of LAM-deficient cells near septa and new poles are independent of RipA-mediated cell separation. Additionally, we showed that the multiseptation defect in large LAM-producing cells was qualitatively different from the multiseptation defect in *ripAB* KD cells. The less regular interseptal distances of the *mptA* OE strain suggest that LAM may be involved in proper septal placement or cell elongation immediately after septal placement.

Intriguingly, *ponA1* overexpression rescued Δ*mptA* even though we showed that RipA is not the cause of cell shape defects in Δ*mptA*. Apart from RipA, PonA1 is known to physically interact with another divisome protein LamA, which is involved in maintaining division asymmetry and drug resistance[55]. PonA1 also interacts with the potential scaffold protein MSMEG_1285 (Rv0613c), which is known to interact physically with additional cell envelope proteins[56]. Since PonA1 is involved in both division and elongation, which are processes likely driven by large multi-protein divisome and elongasome complexes, PonA1 is likely to interact with additional proteins involved in growth and division (see reviews[57,58]). These additional interactions and/or its own peptidoglycan biosynthetic activities may explain the RipA-independent rescue of Δ*mptA* with PonA1. Precise molecular mechanisms by which LAM governs the functions of mycobacterial divisome in a RipA-independent manner is an important topic of future research.

The role of LM remains unknown. The LM precursor accumulating in Δ*mptA* disappeared when the mutant was complemented by *mptA*, and this effect was not dependent on the catalytic activity of MptA as it disappeared even in Δ*mptA* complemented with catalytically inactive MptA (see Fig. 1). When we overexpressed MptA in the WT background, even a leaky expression in the absence of acetamide was enough to make mature LM disappear (see Fig. 5). We do not know the molecular mechanism of these phenomena, but we previously proposed that the balance of MptA and MptC is important for the mannan chain termination and LM accumulation[16]. The action of MptC to add α1-2 mannose side chain may be critical for LM accumulation, and MptA, when expressed at a high level, may impose a dominant negative effect on MptC and block its action, resulting in no LM production. Since LM precursor was absent when catalytically dead MptA was produced in Δ*mptA*, LM precursor accumulation may also be dependent on MptC and the dominant negative action of MptA on MptC may not require its catalytic activity. Alternatively, it is possible that MptA imposes a dominant negative effect on the mannosyltransferase that elongates PIM4 to the LM precursor. These mannosyltransferases likely function in a finely coordinated manner to synthesize LM and LAM, requiring balanced regulation of one another. Importantly, the growth and morphological defects of Δ*mptA* were restored by complementation with WT *mptA* even though we could not restore the LM level in these complemented strains, indicating that LM is not the culprit of these phenotypes we focused on in the current study.

We reproduced similar morphological defects in *M. tuberculosis* by either knocking down *mptA* or overexpressing *mptC*. The morphological defects provide support for the observed growth delay of *M. tuberculosis* upon CRISPRi depletion of *mptA*. Furthermore, our data corroborate previous genome-wide predictions that *mptA* is an essential gene in *M. tuberculosis*[34–37]. Nevertheless, our studies on *M. tuberculosis* are limited and further verifications are necessary to clarify the role of lipoglycans in *M. tuberculosis* and other pathogenic mycobacteria. The conditional nature of the growth and morphological defects of *M. smegmatis* Δ*mptA* is intriguing, and it remains possible that the growth defects of the *M. tuberculosis* mutants are also conditional. The molecular mechanism of cell envelope maintenance and the role of LAM that potentially makes *mptA* essential in *M. tuberculosis* remain to be explored.

## Methods

### Growth

For planktonic growth of *M. smegmatis* mc²155, cells were grown in Middlebrook 7H9 supplemented with 15 mM NaCl, 0.2% (w/v) glucose, and 0.05% (v/v) Tween-80 or LB supplemented with 0.05% (v/v) Tween-80 and incubated at 37 °C and 130 rpm until mid-log phase ($OD_{600}$ = 0.5–1.0). For pellicle biofilm growth of *M. smegmatis*, 20 μL planktonic stationary phase culture was diluted 1:100 into 2 mL M63 medium in a 12-well plate and incubated for 3–5 days at 37 °C without shaking. Spent culture medium was filtered through a 0.22 μm filter to remove residual cells. Example macro-images of pellicle morphology were photographed with an iPhone 12 camera (Apple). For colony growth, serial dilutions of planktonic culture were spread evenly over LB or Middlebrook 7H10 (supplemented with 15 mM NaCl and 0.2% (w/v) glucose) solid agar medium and incubated at 30 °C or 37 °C for 3–4 days. Colonies were photographed using Amersham ImageQuant 800 (Cytiva) and colony area was determined using ImageJ software[59]. Example macro-images of colony size and morphology were photographed with an XT-1 mirrorless camera (Fujifilm). *M. tuberculosis* mc²6230 (a ΔRD1/Δ*panCD* attenuated strain[60] was grown in Middlebrook 7H9 supplemented with Middlebrook OADC supplement, 25 μg/mL D-pantothenic acid, and 0.05% (v/v) Tween-80 at 37 °C with shaking.

### Construction of Δ*mptA* and complemented strains

To generate an unmarked *mptA* deletion mutant, genomic regions upstream and downstream of *MSMEG_4241* (*mptA*) were amplified by PCR using primers (5′- GACAGGACTCTAGCCAAAGAACATCGGTCC GGTGTACG −3′ and 5′- CGGCTCGCCGTCGTGGCCTAGGTGTGGAC TGTCGAGCC −3′ for the upstream region and 5′- TAGGCCACGAC GGCGAGC −3′ and 5′- GCTGTCAAACCTGCCAACTTATCACGCTGGTG-GAAGTGAT −3′ for the downstream region) and assembled using NEBuilder HiFi DNA Assembly cloning kit (NEB) into pCOM1[61]. This construct, designated pMUM241, was electroporated into WT *M. smegmatis* and clones that had incorporated the vector into the genome via single homologous recombination event were selected for on Middlebrook 7H10 plates containing 100 μg/mL hygromycin. A "single cross-over" mutant was isolated and grown to an OD of 1 in the absence of hygromycin to allow a second recombination event to complete the gene deletion and excise the vector backbone. The "double cross-over" strain was selected for on Middlebrook 7H10 plates containing 5% sucrose. Candidate clones were confirmed to be sensitive to hygromycin and resistant to sucrose indicating the loss of the vector backbone. The absence of *MSMEG_4241* (*mptA*) was confirmed by PCR with primers (5′- AGTACCTGCGCGAACGTC −3′ and 5′- TGAGCAGTTC-GAAGGTCAGG −3′) using the genomic DNA of the mutant as a template. To generate complemented strains of Δ*mptA*, we first created an expression vector of *mptA* driven by a native promoter. The *mptA* gene (*MSMEG_4241*) is in frame with and directly downstream of *MSMEG_4240*, a putative geranylgeranyl pyrophosphate synthetase gene, suggesting that these two genes form an operon. Ribosomal profiling of this locus also supports co-transcription of these two genes[62]. Thus, in an attempt to complement expression of *mptA* under control of its native promoter, we cloned the 153 bp sequence upstream of *MSMEG_4240* by PCR using 5′- TCCAGCTGCAGAATTCGAC CGAAACGTGACGGCGA −3′ and 5′- GCTCAGCGCCCGCCCCTTTC −3′ as primers. The full *mptA* (*MSMEG_4241*) sequence was PCR-amplified by 5′- AAGGGGCGGGCGCTGAGCATGACACCGACGGAAACCCA −3′ and 5′-ACTACGTCGACATCGATATCATTGACGGCTCGCCGTCG −3′. Both of these PCR fragments were ligated via HiFi cloning into pMV306 digested with HindIII, resulting in pMUM364. To make an expression vector to produce a catalytically inactive D129A mutant of MptA, site-directed mutagenesis was performed using Platinum SuperFi DNA polymerase (Invitrogen) according to kit instructions. The sense and anti-sense primers, 5′- CAGCCGTGCCACGTACTCCTACCTGG −3′ and

5′- GAGTACGTGGCACGGCTGAACAGCGG −3′, were used to mutagenize the aspartic acid residue within the catalytic loop between the third and fourth transmembrane domains[13]. The expression vectors were electroporated and integrated into the mycobacteriophage L5 attachment (attB) site. Transformants were selected for on Middlebrook 7H10 plates containing 50 µg/mL kanamycin.

### Protein analysis by SDS-PAGE and immunoblotting
Cell lysates were prepared by bead-beating as described previously[63]. Cell lysates and culture filtrates were separated by SDS-PAGE (10% or 12% gel). For culture filtrate, an equal volume of each sample was loaded. For cell lysates, an equal amount of protein was loaded to each lane. For immunoblotting, the gel was transferred to a polyvinylidene difluoride (PVDF) membrane (Bio-Rad) in a transfer buffer (25 mM Tris-HCl (pH 8.3), 192 mM glycine, 0.1% SDS (w/v)) at 14 V for overnight. The membrane was blocked in a blocking buffer (5% skim milk in PBS + 0.05% Tween-20 (PBST)) for 1–2 h at room temperature probed with rabbit anti-*M. smegmatis* Mpa (1:2000 dilution), rabbit anti-*M. smegmatis* MptA (1:1000 dilution)[16], rabbit anti-*M. tuberculosis* MptA (1:1000 dilution) or anti-*M. tuberculosis* MptC (1:1000 dilution)[22] in the blocking buffer for 1 h at 4 °C, and washed 10 min 3 times in PBST before being incubated with horseradish peroxidase-conjugated anti-rabbit IgG (Cytiva, Cat# NA9340V) diluted 1:2000 in the blocking buffer for 1 h at room temperature. Finally, the membrane was washed 10 min 3 times in PBST and developed with homemade ECL reagent (2.24 mM luminol, 0.43 mM *p*-coumaric acid, and 0.0036% $H_2O_2$ in 100 mM Tris-HCl (pH 9.35)) and imaged the chemiluminescence using either an ImageQuant LAS-4000mini Imaging System or Amersham ImageQuant 800 (Cytiva). Anti-*M. tuberculosis* MptA antibody was raised in rabbit against a mixture of two peptides (CSPDRRGVQAATPVVNTP and MTTPSHAPAVDLATAKDC) (Medical and Biological Laboratories, Tokyo, Japan) and affinity-purified using the same peptides.

### Lipoglycan extraction and analysis
Pellets from log phase planktonic cultures were harvested and delipidated with chloroform/methanol/water as previously described[22,63]. Delipidated pellets were resuspended and incubated in phenol/water (1:1) at 55 °C for 2 h to extract LM and LAM. The aqueous phase containing lipoglycan was washed with an equal volume of chloroform. The resulting aqueous phase was then concentrated by vacuum centrifugation and resuspended in water to obtain purified LM and LAM. LM/LAM samples were incubated with 0.1 mg/mL Proteinase K for 1 h at 50 °C to remove residual proteins and separated by SDS-PAGE (15% gel) with a constant 130 V voltage using a Bio-Rad system. Lipoglycans on the separated gel were then stained using the ProQ Emerald 488 glycan staining kit (Life Technologies) and visualized using Amersham ImageQuant 800 (Cytiva) as previously described[11,16].

### Live/Dead staining of microcolonies
For microcolony growth, cells from frozen stock were incubated for 48–72 h in liquid LB media supplemented with 0.05% Tween-80 under shaken conditions at 37 °C. The culture was diluted 100x in fresh LB media (without Tween-80) and deposited in the center of a glass-bottomed well in a 96-well plate (MatTek P96G-1.5-5-F). The culture was then grown at 37 °C under static conditions for approximately 48 h. The medium was replaced with fresh LB media supplemented with 0.1% SYTO9 and 0.1% propidium iodide (Invitrogen) and incubated for 1 h. The microcolonies were imaged using a confocal spinning disk unit (Yokogawa CSU-W1), mounted on a Nikon Eclipse Ti2 microscope with a 100x silicone oil immersion objective (N.A. = 1.35), and captured by a Photometrics Prime BSI CMOS camera.

### Microscopy and quantification of cell morphology
Log phase cells were dispensed onto an agar pad (1% agar in water) on a slide glass and imaged at 1000x magnification (100x objective lens,

N.A. = 1.30) with a Nikon Eclipse E600 fluorescence microscope. Coordinates of cell outlines were obtained by analysis of phase contrast micrographs analyzed with Oufti (version 1)[64]. These coordinates were then processed using Python (version 2.7.16) and Python package NumPy (version 1.8.0rc1) with an original python script (https://github.com/IanLairdSparks/Sparks_2023)[65] to extract cell width profiles and maximum cell width values for each cell. Cell width profiles of individual cells were compiled to obtain population level cell width profiles. Graphs were created using either matplotlib (version1.3.1) operated in Python or ggplot2 (version 3.3.3) operated in R (version 4.0.3). All experiments are repeated at least twice, and representative results are shown.

### Determination of MIC
In a 96-well plate, antibiotics were serially diluted 1:2 in wells containing growth medium (Middlebrook 7H9, LB, or LB supplemented with 500 mM sorbitol) containing 50 µg/mL sulbactam and inoculated with *M. smegmatis* culture to an $OD_{600}$ of 0.03. The plate was incubated in biological triplicate for 24 h at 37 °C before the addition of resazurin to a final concentration of 0.0015% (w/v). The plate was incubated for an additional 8 h at 37 °C after which the MIC was determined spectroscopically at 570 and 600 nm, as described previously[66].

### Construction of Δ*embC* strain
To generate an unmarked *embC* deletion mutant, genomic regions upstream and downstream of *MSMEG_6387* (*embC*) were amplified by PCR using primers (5′- GACAGGACTCTAGCCAAAGATACCTACTGG CCGCGATG −3′ and 5′- GTCCGGGTACCACGGCGTCAGGGCTTC-GATGCTAACGG −3′ for the upstream region and 5′- TGACGCCGTGG-TACCCG −3′ and 5′- GCTGTCAAACCTGCCAACTTATGAGCAGGCC GCCGAT −3′ for the downstream region) and assembled via HiFi cloning into pCOM1[61]. This construct, designated pMUM274, was electroporated into WT *M. smegmatis* and clones that had incorporated the vector into the genome via single homologous recombination event were selected for on Middlebrook 7H10 plates containing 100 µg/mL hygromycin. A "single cross-over" mutant was isolated and grown to an OD of 1 in the absence of hygromycin to allow a second recombination event to complete the gene deletion and excise the vector backbone. The "double cross-over" candidates were selected for on Middlebrook 7H10 plates containing 5% sucrose. These clones were confirmed to be sensitive to hygromycin and resistant to sucrose, indicating the loss of the vector backbone, which can be either a WT revertant or Δ*embC*. We used four primers to test the mutants: #1, 5′-TGCGGTGTTCGACGATCC −3′, #2, 5′- GGGACCCTGCGTGAGGC −3′; #3, 5′- CCCCGTCATGGAGCAGC −3′; and #4, 5′- GAGGCTCGATGG-TATGCGAC −3′. The primer pair #1 and 4 anneals to the sequences outside of the upstream and downstream genome regions present in pMUM274 to test the deletion of *embC*. The primers #2 and 3 bind to the vector backbone of pMUM274, and when combined with #1 and 4 respectively, they detect single cross-over mutants. Of the 12 candidates we tested, we found no *embC* deletion mutants. We further tested 6 additional candidates directly by LM/LAM analysis, and all of them showed the WT phenotype.

### Construction of *embC* and *ripAB* CRISPRi KD strains
ATC-inducible CRISPRi knockdown L5 integration vectors expressing guide RNAs targeting *embC* and *ripAB* were obtained from the Mycobacterial Systems Resource[67] and electroporated into WT or Δ*mptA M. smegmatis*. Transformants were selected for on Middlebrook 7H10 plates containing 20 µg/mL kanamycin and knockdown was induced with 50 ng/mL ATC for 8 h (*ripAB* KD) or 20 h (*embC* KD).

### Bioorthogonal labeling and quantification
To label short-term peptidoglycan remodeling, RADA was added to log phase cells at a concentration of 10 µM and incubated for 15 min at

37 °C. Cells were washed twice with LB and imaged by phase contrast and fluorescence microscopy. To quantify RADA fluorescence, non-septated cells were first outlined in Oufti from phase contrast micrographs. Fluorescence profiles from corresponding fluorescence micrographs were mapped to each cell and oriented from dim pole to bright pole. Cell fluorescence profile data were extracted from Matlab files output by Oufti using Matlab (version R2022b) and analyzed using a custom python script. The highest values for each cell's fluorescence profile were normalized to 1 and a high-order polynomial regression curve was fitted to the population-level data to produce an average fluorescence profile for each strain. To fully label cell walls and septa, HADA was added to log phase cells at a concentration of 500 μM and incubated for 1 h at 37 °C. Cells were washed twice with LB and imaged by phase contrast and fluorescence microscopy. The number of septa per cell was manually counted. The distances between septa in multi-septated cells were measured using ImageJ. All experiments are repeated at least twice, and representative results are shown.

### ponA1 overexpression vector

The WT *ponA1* gene (*MSMEG_6900*) was PCR-amplified from *M. smegmatis* genomic DNA using two primers (5'- AAAAAAAACA-TATGAATAACGAAGGGCGCCACTCC −3' and 5'- TTTTTTATCGATT-CACGGAGGCGGCGGG −3'), which contain NdeI and ClaI sites (underlined), respectively. The PCR product was digested with NdeI and ClaI, and ligated into pMUM261, which was digested with the same enzymes. pMUM261 is a variant of pMUM110, an integrative expression vector driven by the weak P766-8G promoter[68]. Specifically, in pMUM261, the 5'-UTR up to the HindIII site was replaced with 5'-AAGCTTTTTGGTATCATGGGGACCGCAAAGAAGAGGGGCATATG −3' to remove an NcoI site and introduce an NdeI site at the start codon, and additionally a 54 bp fragment containing multiple restriction enzyme sites (HpaI-ClaI-AflII-NheI-NcoI) located at the downstream of the TetR38 gene was removed by digesting with NcoI and HpaI, blunt-ending, and self-ligating. Because the P766-8G-driven expression of the resultant pMUM303 plasmid was too low to rescue ΔmptA, we then swapped in the promoter region of pMUM100 (a variant of pMUM106[68], but carrying the strong P750 promoter) by digesting both with NheI and HindIII and ligating the relevant fragments, resulting in pMUM314, a P750-driven PonA1 expression vector. The plasmid was electroporated and selected with 50 μg/mL streptomycin.

### Catalytically inactive *M. tuberculosis mptC* overexpression vector

We followed a previously described protocol[11]. Briefly, we used pYAB230, a previously published *mptC* (Rv2181) overexpression vector[16], as a template. The site-directed mutation of D46A was introduced by Platinum Pfx DNA polymerase (Invitrogen), using 5'-CGCCGTATCGCATCGCGATCGACATCTATCAG −3' and 5'- CTGATA-GATGTCGATCGCGATGCGATACGGCG −3', as sense and anti-sense primers, respectively, resulting in pYAB297. D46A is equivalent of D45A mutation in the *M. smegmatis* ortholog MSMEG_4247[16]. *M. tuberculosis* mc²6230 was transformed with pYAB230, pYAB297, as well as an empty vector by electroporation, and selected on 7H10 plates supplemented with 100 μg/mL hygromycin.

### Bocillin labeling

PonA1 was detected by Bocillin™ FL (Thermo Fisher) as described[69]. Briefly, cell lysate containing 30 μg protein was incubated with 100 pmol of Bocillin™ FL in a total volume of 7.5 μl for 30 min at 37 °C. The sample was mixed with 2.5 μl of 4x SDS-PAGE loading dye and incubated for 3 min at 98 °C. The entire sample was then analyzed using SDS-PAGE, and PonA1 bound by Bocillin was visualized utilizing the Amhersham ImageQuant 800 (Cytiva).

### Acetamide-inducible MptA overexpression

WT *M. smegmatis* was transfected with either pJAM2 empty vector or pYAB262, a previously published pJAM2-based acetamide-inducible vector to overexpress *mptA*[16]. The transformed strains were grown in Middlebrook 7H9 containing 20 μg/mL kanamycin and *mptA* over-expression was induced with 0.2% acetamide for 8 h.

### CRISPRi knockdown of *M. tuberculosis mptA*

Published protocols were followed with specific modifications[37,70]. A single-guide RNA (sgRNA) was designed using a web-based tool (https://pebble.rockefeller.edu/tools/sgrna-design/). Two oligonu-cleotides (5'-GGGAGCTGTAGCCAAATCAACCGCT −3' and 5'- AAA-CAGCGGTTGATTTGGCTACAGC −3') were annealed by heating to 95 °C and then cooling gradually at the rate of −0.1 °C/s to 25 °C. The double-stranded sgRNA was ligated into a BsmBI-digested plasmid backbone (pIRL58) using T4 DNA ligase (New England Biolabs). The successful integration of the sgRNA into the plasmid was confirmed by Sanger sequencing. *M. tuberculosis* mc²6230 was transformed with the plasmid by electroporation, and selected on 7H10 plates supple-mented with 50 μg/mL kanamycin. *M. tuberculosis mptA* KD strain was grown in Middlebrook 7H9 medium with and without 100 ng/mL ATC at 37 °C with shaking until stationary phase ($OD_{600} > 1.5$). Cells were then sub-cultured (1:50 dilution, 2° culture) into a fresh medium with and without 100 ng/mL ATC, grown for 12 days until the uninduced culture (no ATC) reached log phase ($OD_{600}$ between 0.4 and 1.0). Cells were sub-cultured one additional time (1:50 dilution, 3° culture) into Middlebrook 7H9 medium with or without 100 ng/mL ATC and grown for 7 days when the uninduced culture (no ATC) reached log phase again. Cells were imaged immediately prior to the third sub-culturing and at the end of the third sub-culturing. Cell pellets from the last subculture (3° culture) were collected for cell lysate preparation and lipoglycan extraction.

### Statistics

We followed standard guidelines to use the student's T-test to compare the means between two groups or ANOVA to compare the means among three or more groups[71]. Thus, statistical significance for all pairwise comparisons was determined by one-tailed student's T-tests, calculated by the "T-Test Calculator for 2 Independent Means" tool (www.socscistatistics.com) and independently confirmed in Excel (Microsoft, version 16.77.1). For comparisons of more than two means, statistical significance was determined by ANOVA and Tukey HSD post-hoc test, calculated by the "One-way ANOVA (Analysis Of Variance) with post-hoc Tukey HSD (Honestly Significant Difference) Test Cal-culator for comparing multiple treatments" tool (astatsa.com, 2016 Navendu Vasavada) (see Supplementary Dataset 1 for statistical para-meters). To compare categorical variables (non-septated cells vs. septated cells) between groups, significance was determined by the chi-square test using the "Chi-Square Calculator" tool (www.socscistatistics.com). To determine the strength of correlation, Pear-son correlation analysis was conducted using the "Pearson Correlation Coefficient Calculator" tool (www.socscistatistics.com), followed by a two-tailed student's T-test using Excel to determine the significance of the correlation.

### Strain and plasmid availability

All mutant strains of *M. smegmatis* and *M. tuberculosis* as well as plas-mids generated in this study are available from the authors upon reasonable request.

### Reporting summary

Further information on research design is available in the Nature Portfolio Reporting Summary linked to this article.

## Data availability

Source data are provided with this paper. The data shown in tables, dot plots, scatter plots, and line plots in main and Supplementary Figs. are provided in the Source Data file. All original gel/blot images are also available in the Source Data file. Raw image data for Fig. 1f can be accessed through Dryad (https://doi.org/10.5061/dryad.1vhhmgr1w)[72]. All other image data used to generate data in this study are available from GitHub under https://github.com/IanLairdSparks/Sparks_2023. Source data are provided with this paper.

## Code availability

Computer code is available from GitHub under https://github.com/IanLairdSparks/Sparks_2023[65].

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

## Acknowledgements

This work was supported by NIH (R21AI168791 to YSM and DP2GM146253 to JY). MP was supported by a fellowship from the University of Massachusetts Amherst as part of the Chemistry-Biology Interface Training Program (National Research Service Award T32 GM008515 and GM139789). We thank Dr. Sloan Siegrist (University of Massachusetts Amherst) for sharing her cell wall labeling expertise and reagents, Dr. Keith Derbyshire (Wadsworth Center) for CRISPRi constructs, Dr. Heran Darwin (New York University) for anti-Mpa antibody, and Dr. William Jacobs (Albert Einstein College of Medicine) for *M. tuberculosis* mc²6230 strain. We thank Stevens Bontemps and Claire Kitzmiller for their help with the experiments.

## Author contributions

I.L.S. and Y.S.M. contributed to the conception, experimental design, execution of experiments, drafting, and editing of this work; T.K., M.P., J.N., and J.Y. contributed to the experimental design, execution of experiments, and editing of this work.

## Competing interests

The authors declare no competing interests.

## Additional information

**Peer review information** : *Nature Communications* thanks the anonymous reviewers for their contribution to the peer review of this work. A peer review file is available.

