## [Peer Review File · Nature Communications]

Lipoarabinomannan mediates localized cell wall integrity during division in mycobacteriaReviewer #1 (Remarks to the Author):

Sparks and collaborators report on *Mycobacterium smegmatis* (Msmg) mutants with various defects in the synthesis of the arabinan portion of LAM which present cell envelope deformations in the vicinity of the septa and new poles of the bacilli. These deformations are associated with the displacement of peptidoglycan biosynthesis. The deformations observed are medium-specific and tend to disappear upon addition of osmoprotectants.

Although the deletion of the *mptA* gene (and phenotypically related overexpression of *mptC*) causes marked and interesting phenotypes in the cell envelope structure and integrity of Msmg, it is not possible to say from the data presented whether these phenotypes are in fact the result of deficient LAM biosynthesis or that of perturbations of a multiprotein complex coordinating cell envelope biosynthesis with cell elongation/division caused by the loss of the MptA protein (or overproduction of *mptC*). Indeed, the fact that cell envelope biosynthesis (including LAM biosynthesis) and cell division/elongation are tightly coordinated through a complex network of interacting proteins makes it difficult to disentangle whether the end product of a cell envelope biosynthetic pathway is actually the cause of a particular phenotype, or whether the phenotype may result from collateral/secondary effects of losing a member of the protein interactome. One way the question can potentially be resolved is by complementing the deletion mutant with an inactive variant of the enzyme (like the authors did with *ponA1* and *M. tuberculosis mptC*); i.e., if an inactive variant of MptA restores a WT phenotype, then one may conclude that the MptA protein, rather than LAM, is important for Msmg to maintain its cell envelope structure. To be able to draw accurate conclusions from these experiments, it is also important to verify that the wild-type and mutated variants of the test proteins are expressed at the same level which wasn't done here with *ponA1* and *Mtb mptC* (Fig. 4 and Fig. S6).

Fig. 4: Does the overexpression of active and inactive variants of *ponA1* restore normal LAM synthesis in the *mptA* KO? Otherwise, the results presented in Fig. 4 may actually suggest that the assembly of a functional multiprotein complex coupling cell division with envelope synthesis (somehow restored by *ponA1* overexpression) rather than normal LAM synthesis accounts for the complementation of the mutant's deformation phenotype.

mptA genetic complementation is missing from the experiments presented in Figures 1, S1, S2, S3, Tables S1 and S2, 2, S4, 4, 6 and Table 1.

The SDS-PAGE gel shown in Fig. 3a does not actually provide evidence that the *mptA* KO complemented with *mpta-d2-flag* produces a larger LAM. LM and LAM migrate as heterogeneous glycan populations on SDS-PAGE. Their migration is impacted not only by their size but also by their degree of acylation, the structure of their mannan and arabinan domains and their charge. Detailed structural analyses are required to determine if and how the LAM from the *mptA* KO complemented with *mpta-d2-flag* differs from that of WT Msmg.

Other comments

- Are other cell envelope biosynthetic machineries than that of peptidoglycan displaced in the *mptA* KO mutant (e.g., biosynthesis of mycolic acids and mycolic acid-containing lipids)? In other words, can the observed distortion specifically be associated with displaced peptidoglycan synthesis? This assumption seems to be essentially based on increased susceptibility to beta-lactams.
- Does (active and inactive) *ponA1* overexpression restore the localization of peptidoglycan synthesis (RADA labeling as shown in Fig. S4)?
- Fig. 3: The silencing of the *embC* gene does not result in the expected lipoglycan profile in that no build-up of a truncated LAM is observed. Since *embC* is not an essential gene in *M. smegmatis*, repeating the experiment with a null *embC* KO would provide a much cleaner assessment of the impact of LAM synthesis on the phenotype of interest.
- Fig. 3 panels b and c are missing the uninduced *embC* conditional knockdown control.
- Fig. S5: The conclusion from this experiment should be derived from the analysis of a large number of bacilli and statistics should be provided.

Reviewer #2 (Remarks to the Author):

Sparks et al. investigated how LM/LAM determines growth, separation, and division behaviors in *M. smegmatis*, with confirmatory assays in *M. tuberculosis*. By creating deletion or overexpression mutants for genes involved in the PIM, LM, and LAM biosynthesis pathway, they determined how various structures at different parts affect cell growth and septation. Using FDAA labeling, they show that most of the growth defects and blebbing occur at the new pole. This work is elegantly done and lays down important groundwork that mycobacteriologists need so they can better understand growth and division behaviors and how they relate to virulence and survival during infection. I have a few suggestions for clarity.

Major comments:

- 1) Pg 8 line 27: The blebbing looks like it is peripolar in the images and not strictly polar – this is also consistent with the new pole/septal localization of the deformations (Page 9). In this section, it should be specified that using FDAAs to identify old vs new poles is contained to *M. smegmatis* (because increased polar growth from the old pole has been well studied compared to *M. tuberculosis*). Even so, it is not clear that one can determine pole age using FDAA labels. I suggest that the conclusions re old pole vs new pole be tempered, or that live-cell imaging be performed to determine pole age.
- 2) pg 9. line 13. The section “MptA-Dendra2-FLAG localizes to cell septa” relies on colocalization patterns of MptA-Dendra2-FLAG with the septa in the deletion strain. The authors should demonstrate that this colocalization is also present in WT cells and is not due to the unexpected partial complementation that is mentioned before (pg 8. line7). This partially complemented strain exhibits multisepta unlike the WT strain.

Minor comments:

- 3) The title is vague and awkwardly worded – consider a new title?
- 4) The results would benefit from more logical transitions between sections.
- 5) References are missing on Page 5, lines 2-4.
- 6) pg 5. line 5. It is not clear what the rationale is for measuring colony growth at different temperatures.
- 7) pg 5. line 25-26. It is mentioned that the pellicle of WT and mutant appearance looks different. Please describe how.
- 8) pg5. line 33-pg6. The results appear to be from figure 2 and not supp figure 2. This is a typo, I think.
- 9) pg6. line 17. “the cell morphology was normal”. Only width (and bulging) was measured; I think this is appropriate for this study, but it may be better to explain why other types of morphological features are not quantified and compared. It is not clear why length and aspect ratios, for example, are not considered.
- 10) Some aspects of the experimental methods should be listed in the main text rather than referring to supplementary materials and published literature. For example, what are the culturing and media conditions used throughout the study? For example, was 7H9 supplemented with detergents and other reagents as is common for *M. smegmatis*?
- 11) pg 11. line14. The mptA OE strain is mentioned several times and compared with other strains re large LAM. It would be nice to see this large LAM in the Western (figure 3a) to be compared with other strains.
- 12) pg 11. line 34. The authors report that the spacing between septa is different in Δ mptA L5::mptA-dendra2-flag and induced WT L5::ripAB KD strains. Is the spacing proportional to cell length? This is interesting to follow up on because the authors mentioned that failed cell separation or a defect in cell elongation leads to multiseptated states. One idea is to report cell length in the two strains.
- 13) In the supp section “biorthogonal labelling and quantification”, no reason is given for treating at such different concentrations of RADA (10uM) and HADA (500uM). Was it confirmed that using 500uM of high concentration of HADA does not affect cell growth?

Reviewer #3 (Remarks to the Author):

The authors report an interesting study on the role of mycobacterial lipoarabinomannan (LAM) in the regulation of septation. LAM and biosynthetically related lipomannan (LM) are lipoglycans produced by mycobacteria and related genera. Their role in the modulation of immune response is well-established and has been thoroughly investigated in the past decades. However surprisingly, little is known about their physiological functions. They are hypothesized to functionally replace lipoteichoic acid otherwise produced by Gram-positive bacteria. Yet, it has not been clearly demonstrated. Here, using a series of *Mycobacterium smegmatis* (a widely used non-pathogenic fast-growing mycobacterial model strain) mutants and recombinant strains with altered lipoglycan content, the authors show that LAM, but not LM, regulates septation and division through modulating cell wall integrity and controlling peptidoglycan dynamics. Findings in *M. smegmatis* are tentatively extended to *Mycobacterium tuberculosis* thanks to the study of an LM/LAM mutant. The work is carefully performed and the manuscript is clearly written. However, some points need to be addressed:

- It was previously reported (Patterson, *Biochem J*, 2003; PMID: 12593673) in *M. smegmatis* that mannose-containing molecules, which include LM and LAM, play a role in regulating septation and cell division. This seminal work should be acknowledged and the reference cited.
- Conclusions clearly raised with *M. smegmatis* are tentatively extended to *Mycobacterium tuberculosis* thanks to the analysis of a single mutant. This might be oversimplified. Indeed, i) the *M. tuberculosis* mptc OE has a mild phenotype in term of LAM/LM content; a mutant with a much clearer phenotype such as embC KD should be investigated; what about an mtpA OE?; ii) arabinan is essential for *M. tuberculosis* viability whereas it is not for *M. smegmatis*; iii) peptidoglycan synthesis shows distinct spatiotemporal dynamics between *M. smegmatis* and *M. tuberculosis* (PMID: 28900018). Conclusions raised with *M. smegmatis* are not directly transposable to *M. tuberculosis*. Since the latter is the clinically relevant species, in depth study with this species is required.
- LAM regulates septation and division through modulating cell wall integrity and controlling peptidoglycan dynamics. What are the molecular mechanisms behind? It would be interesting to determine whether LAM deficiency has an impact on cell envelope ultrastructure.
- Lipoglycans are found in Gram-positive bacteria that lack lipoteichoic acids. Yet, in most of these bacteria, except mycobacteria, lipoglycans do not harbor an arabinan domain and are restricted to either LM or LM substituted with single arabinose units. Therefore, the specific role of arabinan size found in *M. smegmatis* is not generalizable to these bacteria. What are authors' hypotheses?

Minor points

- Fig S1: does Δ mptA show the same phenotype than mtpA KD in term of Mpa, glycans and nucleic acid release in the cytosol?
- Fig 3a: mtpA OE was previously shown by the authors to have increased size of both mannan and arabinan domain (Ref 18). It would be interesting to perform chemical analysis of LAM from Δ mptA L5::mptA dendra2-flag strain to determine whether the same applies. According to LAM structure observed, this might allow determining whether arabinan solely again is responsible for the phenotype observed.

Point-by-point Responses

Reviewer #1 (Remarks to the Author):

Sparks and collaborators report on *Mycobacterium smegmatis* (Msmg) mutants with various defects in the synthesis of the arabinan portion of LAM which present cell envelope deformations in the vicinity of the septa and new poles of the bacilli. These deformations are associated with the displacement of peptidoglycan biosynthesis. The deformations observed are medium-specific and tend to disappear upon addition of osmoprotectants.

Although the deletion of the *mptA* gene (and phenotypically related overexpression of *mptC*) causes marked and interesting phenotypes in the cell envelope structure and integrity of Msmg, it is not possible to say from the data presented whether these phenotypes are in fact the result of deficient LAM biosynthesis or that of perturbations of a multiprotein complex coordinating cell envelope biosynthesis with cell elongation/division caused by the loss of the MptA protein (or overproduction of *mptC*). Indeed, the fact that cell envelope biosynthesis (including LAM biosynthesis) and cell division/elongation are tightly coordinated through a complex network of interacting proteins makes it difficult to disentangle whether the end product of a cell envelope biosynthetic pathway is actually the cause of a particular phenotype, or whether the phenotype may result from collateral/secondary effects of losing a member of the protein interactome. One way the question can potentially be resolved is by complementing the deletion mutant with an inactive variant of the enzyme (like the authors did with *ponA1* and *M. tuberculosis mptC*); i.e., if an inactive variant of MptA restores a WT phenotype, then one may conclude that the MptA protein, rather than LAM, is important for Msmg to maintain its cell envelope structure. To be able to draw accurate conclusions from these experiments, it is also important to verify that the wild-type and mutated variants of the test proteins are expressed at the same level which wasn't done here with *ponA1* and Mtb *mptC* (Fig. 4 and Fig. S6).

We thank the reviewer #1 for their constructive comments.

We have made a catalytically inactive version of *mptA* and demonstrated that wildtype *mptA*, but not catalytically dead *mptA*, restores the $\Delta mptA$ phenotypes (see Fig. 1b-e, 2g-l, S1a-e, S3b-c, Table S3 and Page 5, Lines 10-25; Page 6, Lines 10-11, 15-17; Page 7, Lines 3-5; Page 8, Lines 1-2; Page 9, Lines 18-20). We confirmed that both versions of MptA are produced at similar levels by western blotting (see Fig. 1b and Page 5, Lines 17-20). These data support our original suggestion that LAM rather than the MptA protein is critical for the observed growth and morphological defects of $\Delta mptA$.

Furthermore, as requested, we now provide a western blotting of MptC in Mtb strains overexpressing MptC and showed that the levels of wildtype and mutant MptC are comparable (see Fig. S4d and Page 13, Lines 5-8).

We used Bocillin to demonstrate the overexpression of PonA1 as this fluorescent penicillin has been successfully used to detect penicillin-binding proteins like PonA1 in *M. smegmatis*. We showed by Bocillin labeling that wildtype PonA1 was overexpressed in $\Delta mptA + ponA1$ strain relative to $\Delta mptA$ (Fig. 4b, Page 10, Lines 4-6).

Regarding catalytically inactive versions of PonA1, in response to Reviewer #2, we removed them from the revised manuscript. Reviewer #2 requested to improve the logical progression of the Results section. Since the rescue of the mutant phenotype by catalytically inactive versions of PonA1 was partial, and we subsequently showed that RipA was not the culprit, it was more straightforward not to go into the detailed analysis of the mutant versions of PonA1.

Fig. 4: Does the overexpression of active and inactive variants of *ponA1* restore normal LAM synthesis in the *mptA* KO? Otherwise, the results presented in Fig. 4 may actually suggest that the assembly of a functional multiprotein complex coupling cell division with envelope synthesis (somehow restored by *ponA1* overexpression) rather than normal LAM synthesis accounts for the complementation of the mutant's deformation phenotype.

We did not expect that PonA1 expression would restore the LM/LAM phenotype of $\Delta mptA$. We provided the rationale for this experiment in the original manuscript but revised further to clarify our reasoning in the revised manuscript (see Page 9, Line 28 – Page 10, Line 8).

Furthermore, as mentioned above, our new experiments showed that PonA1 was expressed more in the $\Delta mptA + ponA1$ strain than in $\Delta mptA$ by Bocillin labeling (see Fig. 4b; Page 10, Lines 4-6) but LM/LAM biosynthetic defect of $\Delta mptA$ was not restored (see Fig. 4c; Page 10, Lines 6-8). Together, these data provided reasoning for us to test RipA, a known peptidoglycan hydrolase and interacting partner with PonA1.

As shown in our original manuscript, CRISPRi knockdown of *ripA* did not rescue the morphological defects of $\Delta mptA$, indicating that the rescue of the morphological defects of $\Delta mptA$ by PonA1 is likely through LAM acting on other cell envelope synthesis machinery. We attempted to clarify this point in the Discussion (Page 15, Lines 19-30), and it is one important direction we wish to explore in the future.

mptA genetic complementation is missing from the experiments presented in Figures 1, S1, S2, S3, Tables S1 and S2, 2, S4, 4, 6 and Table 1.

Since we newly created complemented strains of $\Delta mptA$ carrying either wildtype or catalytically inactive version of *mptA*, we compared these two strains to address this comment. In all cases, complementation restored the growth, antibiotic sensitivity, and morphological defects of $\Delta mptA$ only when wildtype *mptA* gene was used. The lack of phenotype restoration by catalytically dead *mptA* reinforce our notion that the lack of LAM rather than the lack of MptA protein is the cause of the $\Delta mptA$ mutant phenotype.

Regarding the original Figure S1, in response to the comment by the Reviewer #2, we removed *mptA* KD data entirely from the revised manuscript to simplify the Results section. We replaced the key results (total protein by silver staining and Mpa western blot) with wildtype, $\Delta mptA$, $\Delta mptA + mptA$, $\Delta mptA +$ catalytically inactive *mptA*. Since the release of proteins was sufficient to make our point in the Figure S1, we also removed glycan staining and nucleic acid staining results.

Similarly, in response to Reviewer #2's comment, we removed panels D-E, which used *mptA* KD strains, from the original Figure S2. The results are largely redundant with the data obtained using planktonic cells and sorbitol (former Fig. S3, which is now Fig. S2). We believe that removing the original Fig. S2d-e made the logical progression of the Results section simpler and more straightforward. Fig. S2a-c is included as a part of Fig. S1 in the revised manuscript.

Fig. 2 was revised as requested. All panels now contain data from complemented strains.

Since we showed that the morphological defect was restored by *mptA* complementation in Fig. 2, we do not believe that complemented strains are critical for the points we made in Fig. S3 (new Fig. S2 in the revised manuscript).

We provided data for complemented strains in Fig. S4 (new Fig. S3 in the revised manuscript), as requested.

Fig. 4 shows the restoration of morphological defects of $\Delta mptA$ by PonA1. Genetic complementation of $\Delta mptA$ was shown in the revised Fig. 1, 2, and S1 (see above).

With regard to the antibiotic sensitivity testing (Tables S1 and S2), we felt that it is important to test the complemented strains when the defects were the most pronounced. Therefore, we tested the complemented strains grown in LB (new Table S3).

There was no Figure 6 or Table 1 in our original manuscript.

The SDS-PAGE gel shown in Fig. 3a does not actually provide evidence that the *mptA* KO complemented with *mptA*-d2-flag produces a larger LAM. LM and LAM migrate as heterogeneous glycan populations on SDS-PAGE. Their migration is impacted not only by their size but also by their degree of acylation, the structure of their mannan and arabinan domains and their charge. Detailed structural analyses are required to determine if and how the LAM from the *mptA* KO complemented with *mptA*-d2-flag differs from that of WT Msmg.

We realize that *MptA* overexpression strain used in Fig. 3 does not add much to the point we were making in this figure. We therefore removed it from Figure 3.

Additionally, the complementation using this construct did not fully restore the $\Delta mptA$ mutant phenotype. Therefore, as the Reviewer #2 pointed out, it is possible that the localization of *MptA*-Dendra2 may not represent the physiological localization. We therefore removed Figure S5 as well. We further replaced Figure 5 with our previously established overexpression system, in which we have shown that *MptA* overexpression leads to larger LAM with larger mannan and arabinan moieties.

Other comments

- Are other cell envelope biosynthetic machineries than that of peptidoglycan displaced in the *mptA* KO mutant (e.g., biosynthesis of mycolic acids and mycolic acid-containing lipids)? In other words, can the observed distortion specifically be associated with displaced peptidoglycan synthesis? This assumption seems to be essentially based on increased susceptibility to beta-lactams.

We do not know if other cell envelope biosynthetic machineries are affected in $\Delta mptA$. Since arabinogalactan and mycolic acid layers are covalently connected to peptidoglycan layer, these additional layers may well be affected in $\Delta mptA$. We added these possibilities in the Discussion (Page 15, Lines 2-6) and indicated that it is an important point to be addressed in the future.

- Does (active and inactive) *ponA1* overexpression restore the localization of peptidoglycan synthesis (RADA labeling as shown in Fig. S4)?

As requested, we have done the RADA labeling of $\Delta mptA$ complemented with *ponA1*. Since we removed the data on catalytically inactive PonA1 in the revised manuscript, we focused on the complemented strain expressing active wildtype *ponA1* (see Fig. 4g; Page 10, Lines 9-11).

- Fig. 3: The silencing of the *embC* gene does not result in the expected lipoglycan profile in that no build-up of a truncated LAM is observed. Since *embC* is not an essential gene in *M. smegmatis*, repeating the experiment with a null *embC* KO would provide a much cleaner assessment of the impact of LAM synthesis on the phenotype of interest.

Since EmbC is the processive arabinosyltransferase that acts on a single arabinose residue that is added to the mannan backbone by an unknown priming arabinosyltransferase, the lack of EmbC results in heavily truncated LAM, which only has one priming arabinose attached to LM. Such a molecule is virtually impossible to resolve from LM by SDS-PAGE. Therefore, “build-up of a truncated LAM” suggested by the Reviewer #1 is not expected in either *embC* CRISPRi knockdown or clean *embC* knockout.

What we observed in the *embC* knockdown (a reduced level of LAM) aligns with our expectations and previously published data (for example, see Figure 2 of Goude *et al.*, 2008, PMID: 18424526). We demonstrated that the CRISPRi-induced reduction in LAM content was enough to induce the morphological defects. Consequently, we believe that conducting further experiments using an *embC* null mutant may not substantially contribute to the main point we are trying to convey.

- Fig. 3 panels b and c are missing the uninduced *embC* conditional knockdown control.

Our revised manuscript now contains the uninduced *embC* knockdown control (Fig. 3).

- Fig. S5: The conclusion from this experiment should be derived from the analysis of a large number of bacilli and statistics should be provided.

As mentioned above, we decided to remove this figure from the manuscript.

Reviewer #2 (Remarks to the Author):

Sparks et al. investigated how LM/LAM determines growth, separation, and division behaviors in *M. smegmatis*, with confirmatory assays in *M. tuberculosis*. By creating deletion or overexpression mutants for genes involved in the PIM, LM, and LAM biosynthesis pathway, they determined how various structures at different parts affect cell growth and septation. Using FDAA labeling, they show that most of the growth defects and blebbing occur at the new pole. This work is elegantly done and lays down important groundwork that mycobacteriologists need so they can better understand growth and division behaviors and how they relate to virulence and survival during infection. I have a few suggestions for clarity.

We are grateful of constructive comments and suggestions by the Reviewer #2.

Major comments:

1) Pg 8 line 27: The blebbing looks like it is peripolar in the images and not strictly polar – this is also consistent with the new pole/septal localization of the deformations (Page 9). In this section, it should be specified that using FDAAs to identify old vs new poles is contained to *M. smegmatis* (because increased polar growth from the old pole has been well studied compared to *M. tuberculosis*). Even so, it is not clear that one can determine pole age using FDAA labels. I suggest that the conclusions re old pole vs new pole be tempered, or that live-cell imaging be performed to determine pole age.

This is a point which has been addressed by Hesper Rego and her colleagues already (PMID: 30324906). We referenced this paper and clarified our rationale for using FDAAs to determine the pole age (Page 9, Lines 4-9).

2) pg 9. line 13. The section “MptA-Dendra2-FLAG localizes to cell septa” relies on colocalization patterns of MptA-Dendra2-FLAG with the septa in the deletion strain. The authors should demonstrate that this colocalization is also present in WT cells and is not due to the unexpected partial complementation that is mentioned before (pg 8. line7). This partially complemented strain exhibits multisepta unlike the WT strain.

Since this construct did not fully complement the $\Delta mptA$ mutant phenotype, we agree with the reviewer’s concern that the localization of this protein may not be physiological. We felt that demonstrating the localization in the wildtype may not resolve this concern. Therefore, we decided to remove all data that used the MptA-Dendra2-FLAG-expressing strain. Where appropriate, we used previously characterized MptA overexpression strain (Figure 5).

Minor comments:

3) The title is vague and awkwardly worded – consider a new title?

As suggested, we have revised the title to “Lipoarabinomannan mediates localized cell wall integrity during division in mycobacteria”.

4) The results would benefit from more logical transitions between sections.

We thank the reviewer for this comment. We have carefully revised the manuscript. Some of the major changes we made are that:

1. We removed MptA-Dendra2-FLAG strain from Figure 3 as it was not essential for the point of the Figure. We also removed other data using this strain in the revised manuscript for the reasons described above.
2. We removed *mptA* KD strain from the revised manuscript as the data obtained from MptA KD was largely redundant with those from $\Delta mptA$.
3. We removed data using catalytically inactive versions of PonA1 because the restoration of $\Delta mptA$ phenotype was partial and we have provided more direct data using *ripA* CRISPRi knockdown.
4. We removed glycan and nucleic acid analyses from Figure S1 as it was largely redundant with the protein release data visualized by silver staining and anti-Mpa western blot.

These changes allowed us to streamline the logical progression and simplified the storyline.

- 5) References are missing on Page 5, lines 2-4.

We have added the references (see Page 5, Lines 4-6).

- 6) pg 5. line 5. It is not clear what the rationale is for measuring colony growth at different temperatures.

The reason was because two previous studies from two different groups used these two different temperatures (30 and 37°C). We wanted to confirm that the observed phenotypic differences of $\Delta mptA$ and *mptA* knockdown reported previously were not due to the growth temperature. In the revised manuscript, we tried to clarify our reasoning better (see Page 4, Line 32 – Page 5, Line 2).

- 7) pg 5. line 25-26. It is mentioned that the pellicle of WT and mutant appearance looks different. Please describe how.

We did not mean that the pellicle appearance of WT and mutant looked different. We have described the phenotypic differences in the revised manuscript (See Page 6, Lines 5-7).

- 8) pg5. line 33-pg6. The results appear to be from figure 2 and not supp figure 2. This is a typo, I think.

We apologize for the confusing flow of the Results section in our original manuscript. In response to the Reviewer 2's request, we have tried to improve the logical flow of our revised manuscript. As mentioned above, we removed the original Fig. S2d-e where we showed data from an *mptA* knockdown strain. The original Fig. S2a-c is now included as a part of Fig. S1 in the revised manuscript, in which we put together the analyses on pellicle-grown cells. Figure 2 remains focused on planktonically grown cells, demonstrating medium-dependent morphological and growth defects of $\Delta mptA$.

- 9) pg6. line 17. "the cell morphology was normal". Only width (and bulging) was measured; I think this is appropriate for this study, but it may be better to explain why other types of morphological features are not quantified and compared. It is not clear why length and aspect ratios, for example, are not considered.

We thank the reviewer for this comment. Our mutants only showed minor changes in the cell length, and we now show these data in Fig. 2l, 3d, 4f, 5d, 6f, S1e, and S4d.

- 10) Some aspects of the experimental methods should be listed in the main text rather than referring to supplementary materials and published literature. For example, what are the culturing and media conditions used throughout the study? For example, was 7H9 supplemented with detergents and other reagents as is common for *M. smegmatis*?

We have moved all method descriptions to the main manuscript.

11) pg 11. line14. The *mptA* OE strain is mentioned several times and compared with other strains re large LAM. It would be nice to see this large LAM in the Western (figure 3a) to be compared with other strains.

As discussed above, concerns associated with the MptA-Dendra2-FLAG construct led us to remove the data generated using this construct. Instead, in the new Figure 5, where an MptA overexpression strain can contribute to the key point of the experiment, we used the previously established expression vector that overexpresses MptA (without a fluorescent protein tag). We have previously shown that overexpression of MptA using this vector induces the production of large LAM with more arabinose and mannose relative to inositol (Sena, 2010; PMID: 20215111). In Figure 5b of the revised manuscript, we provided SDS-PAGE visualization of LAM from this strain with or without acetamide induction side by side with LM and LAM from the wildtype as a reference.

12) pg 11. line 34. The authors report that the spacing between septa is different in $\Delta mptA$ L5::*mptA-dendra2-flag* and induced WT L5::*ripAB* KD strains. Is the spacing proportional to cell length? This is interesting to follow up on because the authors mentioned that failed cell separation or a defect in cell elongation leads to multiseptated states. One idea is to report cell length in the two strains.

We thank the reviewer for this insightful comment. We removed the *mptA-dendra2-flag*-expressing strain from the revised manuscript for the reasons stated above. Therefore, to address the Reviewer's point, we reanalyzed the strain overexpressing *mptA* (instead of *mptA-dendra2-flag*) and compared its multiseptation defect to that of the *ripAB* KD strain in Fig. 5. As suggested, we characterized the relationship between cell length and the spacing between septa for both strains (see Fig. 5i; Page 12, Lines 3-9).

13) In the supp section "biorthogonal labelling and quantification", no reason is given for treating at such different concentrations of RADA (10uM) and HADA (500uM). Was it confirmed that using 500uM of high concentration of HADA does not affect cell growth?

We have been using a higher concentration of HADA for bioorthogonal cell wall labeling in mycobacteria and this concentration does not inhibit the cell growth (PMID: 30198841). A higher concentration is used for HADA to improve the signal-to-noise ratio in the cyan channel, where mycobacteria are known to have a weak diffuse autofluorescence (PMID: 18836064).

Reviewer #3 (Remarks to the Author):

The authors report an interesting study on the role of mycobacterial lipoarabinomannan (LAM) in the regulation of septation. LAM and biosynthetically related lipomannan (LM) are lipoglycans produced by mycobacteria and related genera. Their role in the modulation of immune response is well-established and has been thoroughly investigated in the past decades. However surprisingly, little is known about their physiological functions. They are hypothesized to functionally replace lipoteichoic acid otherwise produced by Gram-positive bacteria. Yet, it has not been clearly demonstrated. Here, using a series of *Mycobacterium smegmatis* (a widely used non-pathogenic fast-growing mycobacterial model strain) mutants and recombinant strains with altered lipoglycan content, the authors show that LAM, but not LM, regulates septation and division through modulating cell wall integrity and controlling peptidoglycan dynamics. Findings in *M. smegmatis* are tentatively extended to *Mycobacterium tuberculosis* thanks to the study of an LM/LAM mutant. The work is carefully performed and the manuscript is clearly written. However, some points need to be addressed:

We thank the Reviewer #3 for their constructive comments and expert suggestions.

- It was previously reported (Patterson, Biochem J, 2003; PMID: 12593673) in *M. smegmatis* that mannose-containing molecules, which include LM and LAM, play a role in regulating septation and cell division. This seminal work should be acknowledged and the reference cited.

We thank the reviewer for this suggestion and apologize for the inadvertent omission of this important paper. It is added in the revised manuscript (Page 4, Lines 11-14).

- Conclusions clearly raised with *M. smegmatis* are tentatively extended to *Mycobacterium tuberculosis* thanks to the analysis of a single mutant. This might be oversimplified. Indeed, i) the *M. tuberculosis* *mptC* OE has a mild phenotype in term of LAM/LM content; a mutant with a much clearer phenotype such as *embC* KD should be investigated; what about an *mtpA* OE?; ii) arabinan is essential for *M. tuberculosis* viability whereas it is not for *M. smegmatis*; iii) peptidoglycan synthesis shows distinct spatiotemporal dynamics between *M. smegmatis* and *M. tuberculosis* (PMID: 28900018). Conclusions raised with *M. smegmatis* are not directly transposable to *M. tuberculosis*. Since the latter is the clinically relevant species, in depth study with this species is required.

As requested, we have provided additional supporting evidence that LAM plays a similar function in *Mtb*. We used *mptA* CRISPRi knockdown to show that *mptA* knockdown results in reduced levels of LM and LAM, and morphological defects (new Fig. 6; Page 12, Lines 13-24). We used *mptA* knockdown rather than *embC* knockdown as it is more complementary to the more extensive *M. smegmatis* data we provided in this study.

We additionally revised the manuscript carefully to indicate that our *Mtb* analysis is not as extensive as the one on *M. smegmatis* (Page 16, Lines 16-25).

- LAM regulates septation and division through modulating cell wall integrity and controlling peptidoglycan dynamics. What are the molecular mechanisms behind? It would be interesting to determine whether LAM deficiency has an impact on cell envelope ultrastructure.

We are very interested in uncovering the molecular mechanisms that govern the role of LAM in septation and division but feel that deciphering the precise molecular mechanisms is beyond the scope of this study. We added additional clarifications in the Discussion on the potential roles of LAM in affecting peptidoglycan layer and other envelope structures (Page 14, Lines 28-30; Page 15, Lines 2-6).

- Lipoglycans are found in Gram-positive bacteria that lack lipoteichoic acids. Yet, in most of these bacteria, except mycobacteria, lipoglycans do not harbor an arabinan domain and are restricted to either LM or LM substituted with single arabinose units. Therefore, the specific role of arabinan size found in *M. smegmatis* is not generalizable to these bacteria. What are authors' hypotheses?

We agree with the Reviewer #3 that arabinan domain cannot explain the general role of lipoglycans in other bacteria. In other bacteria, other parts of the glycan chain may be important, but it is a speculation. We acknowledge that further studies are needed to expand our ideas. We have toned down our discussion in the revised manuscript and emphasized that our hypothesis is a speculation at this time (Page 13, Lines 27-30).

Minor points

- Fig S1: does $\Delta mptA$ show the same phenotype than *mptA* KD in term of Mpa, glycans and nucleic acid release in the cytosol?

In response to the comment by the Reviewer #2, we have removed data obtained using *mptA* KD cells. In the revised manuscript, we therefore provided a new anti-Mpa western blot data using $\Delta mptA$ (Fig. S1b). We removed the data on the release of glycan and nucleic acid as those are largely redundant with the protein release data.

- Fig 3a: *mptA* OE was previously shown by the authors to have increased size of both mannan and arabinan domain (Ref 18). It would be interesting to perform chemical analysis of LAM from $\Delta mptA$ L5::*mptA-dendra2-flag* strain to determine whether the same applies. According to LAM structure observed, this might allow determining whether arabinan solely again is responsible for the phenotype observed.

Because of the concerns raised by other reviewers, we have removed data obtained using the $\Delta mptA$ L5::*mptA-dendra2-flag* strain. As the Reviewer #3 pointed out, we have a previously established construct to overexpress *mptA*, which results in larger mannan and arabinan domains. Therefore, in the revised manuscript, we used the overexpression of native unmodified *mptA* rather than *mptA-dendra2-flag* to address questions associated with Fig. 5 experiments.

Reviewer #1 (Remarks to the Author):

Sparks et al. show that a deficiency in the expression of the alpha-1,6 mannosyltransferase MptA or overexpression of the alpha-1,2 mannosyltransferase MptC, which result in reduced, incomplete or absence of LAM production, alters the rod shape cell morphology of *M. smegmatis* and *M. tuberculosis*. This alteration is characterized by a blebbing of the bacilli that accompanies increased sidewall peptidoglycan (PG) synthesis/remodeling. Conversely, overexpression of mptA in *M. smegmatis*, which is thought to lead to the synthesis of larger size LAM, leads to a hyperseptation phenotype. Collectively, the results point to proper LAM structure being important to the ability of mycobacterial bacilli to divide while preserving localized cell wall integrity.

Critiques raised on the original version of the manuscript were only partially addressed and, thus, some significant concerns remain:

Importantly, the authors now show that genetic complementation with an active version (but not an inactive version) of MptA is required to reverse altered bacilli morphology. Despite this, the manuscript remains very descriptive with little to no mechanistic insights besides the negative results presented in Fig. 4 and 5 concerning the potential involvement of RipA/B in the observed phenotypes. A potentially interesting lead concerns the observations that de novo PG synthesis is increased at the site of cell blebbing and the fact that ponA1 overexpression restores envelope integrity and shape in LM/LAM-deficient mutants. Unfortunately, the authors did not follow up on these observations and the potential impact of LAM on PG synthesis/remodeling. On the contrary, they deleted from the revised manuscript a potentially important observation showing that overexpression of inactive forms of ponA1 may at least partially reverse the phenotypes of LM/LAM-deficient mutants. The only hypothesis explored in the manuscript (and which turns out to be incorrect) is that LAM may be necessary for controlling septal PG hydrolase activity without any clear rationale for privileging it, at least at the point at which this hypothesis is first introduced in the text (p. 9-10).

Given that the point of using an embC mutant is to show that the arabinan domain of LAM is specifically required for normal rod shape cell morphology in *M. smegmatis*, it remains unclear why the authors persist in using a conditional knock-down strain in which LAM production is reduced, instead of an embC clean KO which is completely deficient in the synthesis of the arabinan domain of LAM. embC is not an essential gene in *M. smegmatis* contrary to the situation in *M. tuberculosis*. The Goude et al. citation refers to work performed in *M. tuberculosis* reason why a null mutant couldn't be studied in this case.

Overexpression of mptA leads to an interesting hyperseptation phenotype in *M. smegmatis*. However, nowhere is evidence provided in the manuscript (beyond rather uninformative migration on SDS-PAGE) that the mptA overexpressor actually produces a larger size LAM. Sena et al., 2010 is cited to support this claim but this reference only presents a partial structural analysis of LAM produced by a *M. smegmatis* recombinant strain overexpressing concomitantly mptC and mptA, not mptA alone as is the case in the present study.

Minor comments

Figure 1: Suggest labeling the legend "Proposed pathway for PIM, LM and LAM" given that some of the steps shown in the scheme actually still remain hypothetical.

Figure S1b: What was the protein loading standardized to? This critical detail is missing from the figure legend and Materials and Methods.

It is unclear why different statistical tests (ANOVA or Student's t-test) are used in different figures.

Original, uncropped, blot and gel images are missing from the revised manuscript and, thus, couldn't be reviewed.

Reviewer #2 (Remarks to the Author):

The revision has addressed all of my previous concerns.

Reviewer #3 (Remarks to the Author):

The authors have done extensive work to confirm and better characterize the role played by LAM in the division of *Mycobacterium smegmatis* (and probably *Mycobacterium tuberculosis*), including maintenance of local cell envelope integrity and septal placement. Yet, mechanistic insights are lacking and the conceptual advance is rather limited.

Point-by-point Responses

Reviewer #1 (Remarks to the Author):

Sparks et al. show that a deficiency in the expression of the alpha-1,6 mannosyltransferase MptA or overexpression of the alpha-1,2 mannosyltransferase MptC, which result in reduced, incomplete or absence of LAM production, alters the rod shape cell morphology of *M. smegmatis* and *M. tuberculosis*. This alteration is characterized by a blebbing of the bacilli that accompanies increased sidewall peptidoglycan (PG) synthesis/remodeling. Conversely, overexpression of *mptA* in *M. smegmatis*, which is thought to lead to the synthesis of larger size LAM, leads to a hyperseptation phenotype. Collectively, the results point to proper LAM structure being important to the ability of mycobacterial bacilli to divide while preserving localized cell wall integrity.

Critiques raised on the original version of the manuscript were only partially addressed and, thus, some significant concerns remain:

Importantly, the authors now show that genetic complementation with an active version (but not an inactive version) of MptA is required to reverse altered bacilli morphology. Despite this, the manuscript remains very descriptive with little to no mechanistic insights besides the negative results presented in Fig. 4 and 5 concerning the potential involvement of RipA/B in the observed phenotypes. A potentially interesting lead concerns the observations that *de novo* PG synthesis is increased at the site of cell blebbing and the fact that *ponA1* overexpression restores envelope integrity and shape in LM/LAM-deficient mutants. Unfortunately, the authors did not follow up on these observations and the potential impact of LAM on PG synthesis/remodeling. On the contrary, they deleted from the revised manuscript a potentially important observation showing that overexpression of inactive forms of *ponA1* may at least partially reverse the phenotypes of LM/LAM-deficient mutants. The only hypothesis explored in the manuscript (and which turns out to be incorrect) is that LAM may be necessary for controlling septal PG hydrolase activity without any clear rationale for privileging it, at least at the point at which this hypothesis is first introduced in the text (p. 9-10).

In our previously submitted revised manuscript, we presented our rationale for investigating the potential role of RipA in mediating the rescue of the $\Delta mptA$ phenotype, as outlined on Page 10, Lines 2-8. Our hypothesis was based on a referenced study demonstrating the inhibitory effect of PonA1 on RipA. In our study, PonA1 overexpression successfully rescued the $\Delta mptA$ phenotype, leading us to hypothesize that RipA inhibition plays a role in this rescue.

As Reviewer #1 rightly pointed out, our hypothesis on this specific aspect did not align with our experimental findings, and unfortunately, we were unable to provide additional insights into the molecular mechanism. However, we believe the true value of our study lies in the discovery of LAM as a crucial molecule for maintaining cell wall integrity during cell division.

To illustrate our perspective, we draw a parallel to the Avery–MacLeod–McCarty experiment in 1944, which conclusively identified DNA as the genetic material responsible for bacterial transformation. Notably, the experiment did not elucidate the molecular mechanism by which DNA mediates this transformation. Subsequent studies uncovered more detailed mechanisms, but the initial failure to reveal the molecular intricacies did not diminish the groundbreaking significance of their discovery.

In a similar vein, we acknowledge the limitation of our current study in providing detailed mechanistic insights. Nevertheless, we emphasize that our findings regarding LAM's role in maintaining cell wall integrity represent a valuable contribution. We anticipate that future studies may shed light on the finer mechanisms, but such endeavors are beyond the scope of our present work.

Given that the point of using an *embC* mutant is to show that the arabinan domain of LAM is specifically required for normal rod shape cell morphology in *M. smegmatis*, it remains unclear why the authors persist in using a conditional knock-down strain in which LAM production is reduced, instead of an *embC* clean KO which is completely deficient in the synthesis of the arabinan domain of LAM. *embC* is not an essential gene in *M. smegmatis* contrary to the situation in *M. tuberculosis*. The Goude et al. citation refers to work performed in *M. tuberculosis* reason why a null mutant couldn't be studied in this case.

Despite multiple attempts, we encountered challenges in generating a clean null mutant of *embC*. Regrettably, we currently lack insights into the specific reasons behind our inability to delete *embC* successfully.

In the revised manuscript, we have acknowledged that *embC* has been deleted in previous studies, and yet our attempts to delete the gene was unsuccessful (see Page 20, Lines 11-33).

In light of these challenges, we turned to an alternative approach, employing *embC* CRISPRi knockdown, which yielded the anticipated phenotype. This outcome effectively supports our key point – the crucial role of the arabinan part of LAM in maintaining cell wall integrity.

Given the success and relevance of the CRISPRi knockdown approach, we have determined that obtaining a knockout of the *embC* gene is not indispensable for the core objectives of our study. While we recognize the significance of a clean null mutant, we believe our alternative strategy provides substantial evidence to underscore our research findings.

Overexpression of *mptA* leads to an interesting hyperseptation phenotype in *M. smegmatis*. However, nowhere is evidence provided in the manuscript (beyond rather uninformative migration on SDS-PAGE) that the *mptA* overexpressor actually produces a larger size LAM. Sena et al., 2010 is cited to support this claim but this reference only presents a partial structural analysis of LAM produced by a *M. smegmatis* recombinant strain overexpressing concomitantly *mptC* and *mptA*, not *mptA* alone as is the case in the present study.

We underscore that the key contribution of our present report is the discovery of the physiological function of LAM in mycobacteria.

While our focus is on shedding light on the vital role of LAM, it is crucial to clarify that our primary objective is not to establish a detailed structure-function relationship between the precise LAM structure and peptidoglycan integrity. This level of analysis falls beyond the immediate scope of our current study.

The comprehensive analysis of LAM structure is undoubtedly a labor-intensive process, and even if we were to provide the complete structure, it may not necessarily yield additional mechanistic insights at this stage of our investigation. We believe that our prior reports on the compositional analysis of large LAM, coupled with our current data showcasing LAM with a conspicuously large size on SDS-PAGE, sufficiently support the central point we aim to make.

Our intent is to provide a robust foundation for understanding the physiological function of LAM, paving the way for future studies to delve into more intricate details. We are confident that the significance of our findings lies in the broader implications for mycobacterial biology rather than an exhaustive structural analysis.

Minor comments

Figure 1: Suggest labeling the legend “Proposed pathway for PIM, LM and LAM” given that some of the steps shown in the scheme actually still remain hypothetical.

Revised as suggested (see Page 33, Lines 3-4).

Figure S1b: What was the protein loading standardized to? This critical detail is missing from the figure legend and Materials and Methods.

The protein loading was standardized by loading the same volume of culture filtrates. We indicated this fact in the previously submitted revised manuscript (see Page 18, Line 22).

It is unclear why different statistical tests (ANOVA or Student’s t-test) are used in different figures.

In the previously submitted manuscript, we stated “Statistical significance for all pairwise comparisons was calculated using one-tailed student’s T-tests. For comparisons of more than two means, statistical significance was determined by ANOVA and Tukey HSD post-hoc test”.

To elaborate further, we have additionally provided a reference that explains the reasons why two different statistical tests are used. As stated in the “Basic Concepts” section of the referenced article, “the Student's t test (also called T test) is used to compare the means between two groups and there is no need of multiple comparisons as unique P value is observed, whereas ANOVA is used to compare the means among three or more groups”.

We further provided the online statistical tools used for each statistical method. We hope that the additional descriptions resolve the reviewer’s concern.

Original, uncropped, blot and gel images are missing from the revised manuscript and, thus, couldn’t be reviewed.

We have provided a source data file in Microsoft Excel format, following the journal’s guideline. The source data file contains the raw data underlying all graphs and charts, and uncropped versions of any gels or blots presented in the figures. Within the source data file, each figure panel or table containing relevant data is represented by a single sheet in an Excel document. Following the guideline, blot and gel images are pasted in and labelled with the relevant panel and identifying information such as the antibody used.

Reviewer #2 (Remarks to the Author):

The revision has addressed all of my previous concerns.

Thank you.

Reviewer #3 (Remarks to the Author):

The authors have done extensive work to confirm and better characterize the role played by LAM in the division of *Mycobacterium smegmatis* (and probably *Mycobacterium tuberculosis*), including maintenance of local cell envelope integrity and septal placement. Yet, mechanistic insights are lacking and the conceptual advance is rather limited.

Thank you. Please see our comment to the Reviewer #1 regarding the comments on the mechanistic insights.